

# Intense Anticyclones at the Global Argentine Basin Array of the Ocean Observatory Initiative

Camila Artana[1,2,3] and Christine Provost[1]

[1]Laboratoire LOCEAN-IPSL, Sorbonne Universités (UPMC, University Paris 6), CNRS, IRD, MNHN, Paris, France
[2]Mercator Ocean, Toulouse, France
[3]Institut de Ciènces del Mar, Barcelona, Spain

**Correspondence:** Camila Artana (c.artana@icm.csic.es)

**Abstract.** We analyzed physical oceanic parameters gathered by a mooring array at mesoscale spatial sampling deployed in Argentine Basin within the Ocean Observatory Initiative, a National Science Foundation Major Research Facility. The array was maintained at 42°S 42°W, a historically sparsely sampled region with small ocean variability, during 34 months from March 2015 to January 2018. The data documented four anticyclonic extreme structures events in 2016. The four anticyclonic structures had different characteristics (size, vertical extension, origin, lifetime, Rossby Number). They all featured near-inertial waves (NIWs) trapped at depth and low Richardson values well below the mixed layer. Low Richardson values suggest favorable conditions for mixing. The anticyclonic features likely act as mixing structures at the pycnocline bringing heat and salt from the South Atlantic Central Water to the Antarctic Intermediate Waters. The intense structures were unique in the 29-year-long satellite altimetry record at the mooring site. The Argentine Basin is populated with many anticyclones and mixing associated to trapped NIWs probably plays an important role in setting up the upper water masses characteristics in the Basin.

## 1 Introduction

The Argentine Basin is a very active region of the Southern Ocean with unique oceanographic characteristics and contrasted water masses (Figure 1 a) (Artana et al., 2021). The Malvinas Current (MC), the major western boundary current of the Argentine Basin, flows northward following the Subantarctic Front (SAF) and, at about 38°, encounters the warm and salty Brazil Current. The Brazil Current is the western boundary current of the South Atlantic Subtropical Gyre bounded by the Subtropical Front (STF) (Figure 1 a). The Brazil-Malvinas Confluence is a complex highly dynamic region populated with meso and submesoscale structures such as eddies, rings, filaments, intrusions and meanders (e.g., Orue-Echeverria et al., 2019). The eddy kinetic energy (EKE) features a C-shape pattern with values above 800 x $10^{-4}$ m²/s² that extends from the Brazil-Malvinas Confluence (EKE reaching 1000 x $10^{-4}$ m²/s²) to the Southern Argentine Basin (Figure 1 b). A local minimum of EKE in the center of the Basin (Figure 1 b) corresponds to the Zapiola Anticyclonic Circulation (ZAC, Figure 1 a) (Saraceno et al., 2009). Information on eddies in the Argentine Basin mostly comes from satellite altimetry (e.g. Fu et al., 2006; Saraceno and Provost, 2012; Mason et al., 2017) as in situ time series are scarce in this region. The only mooring data were obtained in the boundary currents either in the Malvinas Current (e.g. Artana et al., 2018) or the Brazil Current (e.g. Meinen et al., 2017) or only at depth near the bottom (e.g. Weatherly, 1993). However, due to the spatial and temporal sampling of present satellite





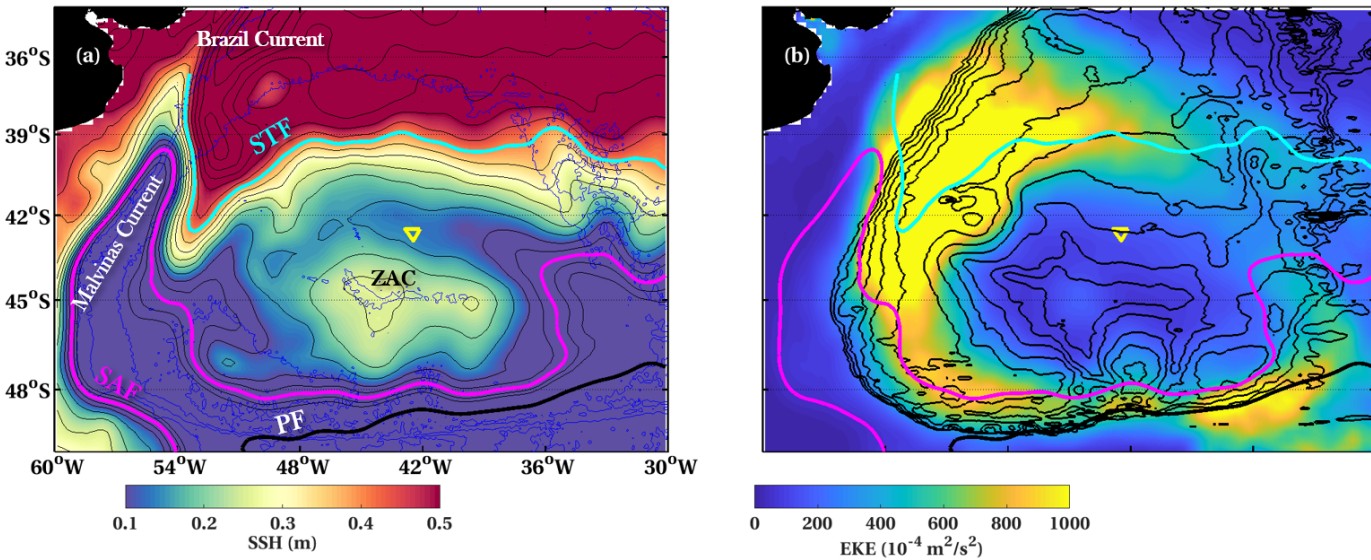

**Figure 1.** Mean sea surface height (SSH) (a) and mean eddy kinetic energy (b) over the period 1993-2020 from satellite altimetry. The cyan, magenta and black contours represent the mean position of the Subtropical Front (STF, SSH=0.40 m), Subantarctic Front (SAF, SSH=0 m) and Polar Front (PF, SSH=0.34 m). The yellow triangle indicates the location of the OOI array. In (a) black isolines are every 5 cm and blue isolines represent isobaths: 6000, 5000, 3000, 2000 from Sandwell and Smith (1994). In (b) black isolines are potential vorticity contours plotted every $5\times10^{-9}$ s$^{-1}$m$^{-1}$. ZAC stands for Zapiola Anticyclonic Circulation and PF for Polar Front.

altimetry missions, only a portion of the mesoscale activity can be monitored as eddies with radius smaller than 40 km are attenuated in satellite altimetric maps (e.g., Chelton et al., 2011; Ballarotta et al., 2019).

     Within the Ocean Observatory Initiative (OOI, http://www.oceanobservatories.org) mooring arrays were deployed to provide sustained open-ocean observations in high-latitude areas that have been historically sparsely sampled due to severe weather conditions (e.g. Smith et al., 2018). These arrays are collecting multidisciplinary data to address science questions related to

ocean-atmosphere exchanges, climate variability, ocean circulation, ecosystems, the global carbon cycle, turbulent mixing and biophysical interactions (e.g., Ogle et al., 2018; Palevsky et al., 2018; Josey et al., 2019). Among the OOI mooring arrays, the Argentine Basin Global array was deployed and maintained at 42°S 42°W during 34 months from March 2015 to January 2018 (Figure 1a). The mooring array was located to the south of the mean position of the Subtropical Gyre and to the north of the Zapiola Anticyclonic Circulation (Figure 1 a), in a region with small westward mean surface geostrophic velocities (-0.02

m/s) and weak mean eddy kinetic energy (145 x $10^{-4}$ m$^2$/s$^2$) (Figure 1). We concentrated on the ocean physical conditions encountered during those three years in this remote and never sampled environment. The array documented unexpected extreme mesoscale structures which are the focus of this work. We aimed at providing elements of response to the following questions. What are these mesoscale structures? Where do they come from? How often do they occur? Do they impact mixing?



The paper is organized as follows. Section 2 presents an overview of the OOI ocean physical data set. Section 3 focuses
on three extreme events recorded by the array. They are identified in satellite altimetry as ocean anticyclonic structures with
distinct characteristics and origins. Section 4 investigates high frequency variations ($< 1$ day) during the events, in particular
near inertial motions and possible occurrence of mixing. Section 5 summarizes and concludes.

## 2   The Global Argentine Array: Overview

### 2.1   Mooring set up

Moorings from the Global Argentine array were deployed for the first time in March 2015 (https://oceanobservatories.org/array/global-
argentine-array/) below Jason ground track #76 and # 35 (Figure 2 a). The ocean depth at the mooring array was 5200 m.
Moorings were recovered and redeployed at the same position in October 2015 and September 2016. January 2018 was the
end of the operations in the Argentine Basin with the final mooring recovery. At each visit, CTD casts with water sampling
were performed at the mooring site for instrument calibration (https://oceanobservatories.org/array/global-argentine-array/).
The array comprised four moorings in a triangular configuration (Smith et al., 2018): the Flanking Mooring A and B were
at the two northern corners of the triangle and the paired Surface and APEX profiler moorings occupied the southern corner
(Figure 1 and 2).

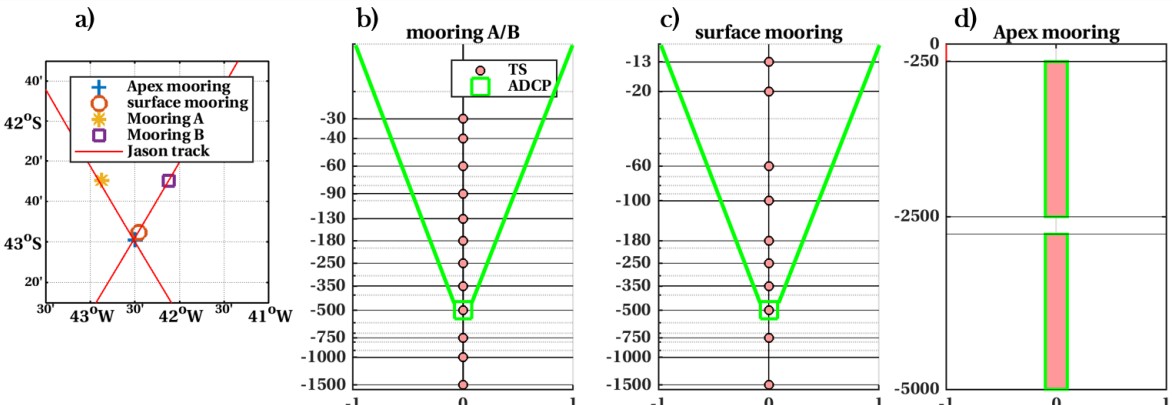

**Figure 2.** a) Location of OOI moorings. Schematics of the northern moorings A and B (b), surface mooring (c) and APEX profiler mooring
(d). The red circles mark the depth of temperature and salinity measurements and the green squares the velocity measurements. Note that the
vertical axis (depth in meters) in b) and c) is in log scale.

The two identical Flanking Moorings A and B (FLMA and FLMB, 62 km apart) comprised 12 microcats fixed at specific
depths between 30 m and 1500 m and an upward looking RDI 75 kHz Long Ranger Acoustic Doppler Current Profiler (ADCP)
located at 500 m measuring velocity profiles (Figure 2 b). The time sampling of the microcats and ADCPs was 15 minutes and
1 hour, respectively (Table 1). The surface mooring (SUMO) had 10 microcats at fixed depths between 13 to 1500 m depth



and an upward looking ADCP RDI (75 kHz Long Ranger) at 500 m (Figure 2 c). The time sampling was 7.5 minutes for the microcats and 1 hour for the ADCP (Table 1). The profiler mooring (APEX), 7 km away from SUMO, was a subsurface mooring equipped with two wire-following McLane moored profilers (Figure 2 d). The profilers performed one to two profiles per day,
continuously sampling ocean variables (in particular temperature, salinity, zonal and meridional velocities) over specified depth intervals (250-2445 m for the upper profiler and 2470-4605 m for the lower one).

| Mooring | $\Delta t$ T/S | $\Delta t$ V |
|---|---|---|
| FLMA/FLMB | 15 min | 1h |
| SUMO | 7.5 min | 1 h |
| APEX profiler | 1-2 profile/day | 1-2 profile/day |

**Table 1.** Time sampling ($\Delta t$) of the mooring instruments.

The data processing of the physical variables is detailed in Artana et al. (2020). We removed spikes from temperature and salinity and density inversions. Measurement quality of the mooring sensors, assessed by comparison with shipboard measurements (CTD profiles and bottle measurements) taken near the moorings during deployment and recovery cruises, was
nominal.

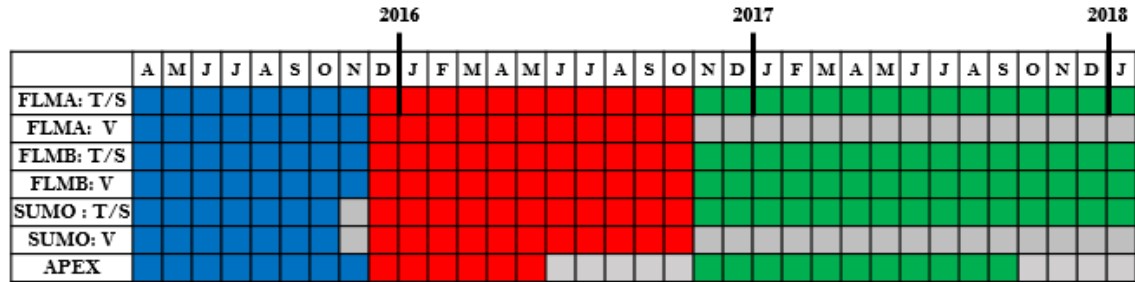

**Table 2.** Record length for deployment 1 (blue), deployment 2 (red) and deployment 3 (green). Grey squares correspond to missing data

ADCP from mooring A (FLMA) and the surface mooring (SUMO) did not work during the third deployment (Table 2). Gaps in the upper layer occurred when FLMB and FLMA underwent vertical excursions due to strong currents. Data quality was overall very good at all depths, except for the velocity measurements close to the surface. Spiky and noisy patterns were often found in winter, concurrent with strong winds sometimes down to 150 m. They were identified and discarded.

## 2.2 Hydrography in the upper 1500 m (2015-2017)

The data return at the SUMO mooring is high in 2016 while data are more incomplete in 2015 and 2017 (Figure 3). The temperature time series in the upper 100 m show a marked seasonal cycle with a range of 10°C at the surface and 6°C at 100



m ( pink and green curves in Figure 3). Waters at 60 m and 100 m depth are saltier than at shallower levels in summer by about 0.11 psu (Figure 3 b). In contrast, as winter mixed layers often exceed 100 m, temperatures and salinities are homogeneous in
the first 100 m and the pink (13 m), red (20 m), dark green (60 m) curves are hidden behind the light green curve (100 m) in Figure 3 from end of May to end of October.

On top of the seasonal cycle, the time series show large variations of week to month duration that extend in the water column. Two temperature (T) and salinity (S) extrema, labelled A1 and A2, exceed three times the standard deviation (in T and S) at 180 m (purple curve in Figure 3 a and b). A1 and A2 happened in austral winter 2016 (Table 3). They were associated with
warm and salty waters (T > 10°C and S > 35 psu). During A2 the instruments below 500 m were lifted up by about 100 m at 1000 m and 200 m at 1500 m depth (brown curves in Figure 3 c). Another remarkable event, labelled A0 occurred around mid-March 2016 and featured the highest recorded surface temperature (T> 17°C at 20 m) (pink curve in Figure 3 and Table 3).

The $\theta$-S diagram from SUMO data documents the water masses occupying the upper 1500 m of the water column (Figure 4). Light densities with $\sigma_\theta$ less than 27.00 kg/m$^3$ correspond to South Atlantic Central Water (SACW) for salinities larger than
34.2 psu and to Subantarctic Surface Water (SASW) for the lower salinities (Maamaatuaiahutapu et al., 1994). The salinity minima in the density range between 27.00 and 27.33 kg/m$^3$ are associated with Antarctic Intermediate Waters located between 200 and 500 m (orange and yellow curves in Figure 3 b). Waters with densities larger than 27.33 kg/m$^3$ in Figure 4 correspond to Upper Circumpolar Deep Water (UCDW). The second deployment data (red in Figure 4) result in distinctive patterns in
the $\theta$-S diagram where the extrema observed in Figure 3 (A0, A1 and A2) stand out. The cloud spread in the $\theta$ - S diagram is considerably reduced when considering daily averaged data instead of full resolution data (7.5 minutes) indicating large high-frequency fluctuations in temperature and salinity (Figures 4 a and b).

FLMA and FLMB did not move much during deployments 1 and 3 while they underwent large draw-downs during deployment 2 (Appendix Figure A1 and Table 3). A0 event led to vertical excursions of 100 m at 1500 m depth at FLMA while FLMB
remained still (Figure A1 from Apendix A1 and Table 3).

Event A1 in May 2016 led to vertical excursions in excess of 400 m at FLMA and FLMB and to the collapse of the APEX mooring (Appendix A1 Figure A1 and Table 3). As a result, the APEX mooring did not sample A1 and A2 events.

| Mooring | A0 | A1 | A2 |
|---|---|---|---|
| FLMB | 15/03-10/04 | draw-downs | 09/07-18/08 |
| FLMA | draw-downs | draw-downs | 15/07-18/08 |
| SUMO | 11/03-10/04 | 27/05-17/06 | 17/07-18/08 |

**Table 3.** Dates of events at each mooring. No date is indicated during strong draw-downs.







**Figure 3.** Temperature (a), salinity (b), and pressure (c) time series at the depths (indicated with colors) sampled at SUMO. The horizontal purple line in a (respectively b) corresponds to the mean plus 3 standard deviations of the temperature (respectively salinity) at 180 m. The color bars in the x-axis delimit the three deployments: deployment 1 (blue), deployment 2 (red) and deployment 3 (green). Three events labelled A0, A1 and A2 stand out.





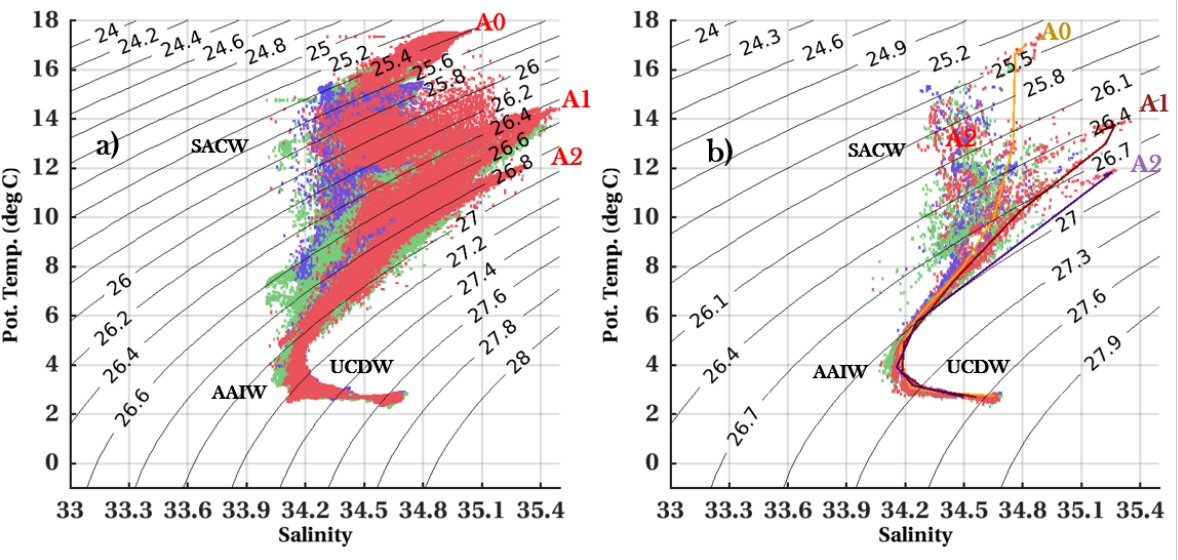

**Figure 4.** a) $\theta$ - S diagram for each deployment from the surface mooring data (sampling every 7.5 minutes). First deployment in blue, second deployment in red and third deployment in green. b) Same as a) with daily averaged data. Daily profiles for April 1, June 9 and August 7 2016 corresponding to A0, A1 and A2 are indicated in b.

## 2.3 Velocities

FLMB ADCP continuously functioned over the three deployments while the ADCP from SUMO and FLMA did not work
during the third deployment (Figure 5, Table 2). The largest velocities (peaking at 0.9 m/s) are observed in 2016 in the time series recorded by the 3 ADCPs (at SUMO, FLMA and FLMB) and are associated with A0, A1 and A2 (Figure 5). To the first order, velocity components were rather homogeneous in the vertical (Figure 5), and statistics for vertically averaged velocities are presented in Table A1 in Appendix A2.

Over the 3 deployments, the vertically averaged mean zonal velocity at FLMB is westward (-0.02 m/s) and the vertically
averaged mean meridional velocity northward (0.01 m/s). The meridional and zonal velocity components of the ADCPs recurrently changed sign leading to small mean values (< 0.04 m/s) and larger standard deviations (>0.08 m/s) (Table A1 from Appendix A2 and Figure 5). The second deployment presents the largest standard deviations and EKE values. The standard deviations reduced by about 1 to 2 cm/s and EKE by 20% when considering daily velocities( Table A1 from Appendix A2).

Vertically averaged velocities between 300 and 480 m from the APEX profiler and the nearby SUMO are correlated (> 0.7
above 99% confidence level). APEX mooring velocities confirm that currents are rather barotropic in the region over the 5000 m depth (Figure 5 g-h).

The surface daily geostrophic velocities derived from satellite altimetry at 1/4° (distributed by Copernicus Marine Service, CMEMS, http://marine.copernicus.eu/) are in good agreement with the vertically averaged velocities at SUMO, FLMA and FLMB in terms of means, standard deviations and EKE (Appendix Table A1). Satellite EKE was small during deployment 1
and 3 (< 100 x $10^{-4}$ m²/s²) and doubled during deployment 2 (> 200 x $10^{-4}$m²/s²) which is consistent with in situ velocities.





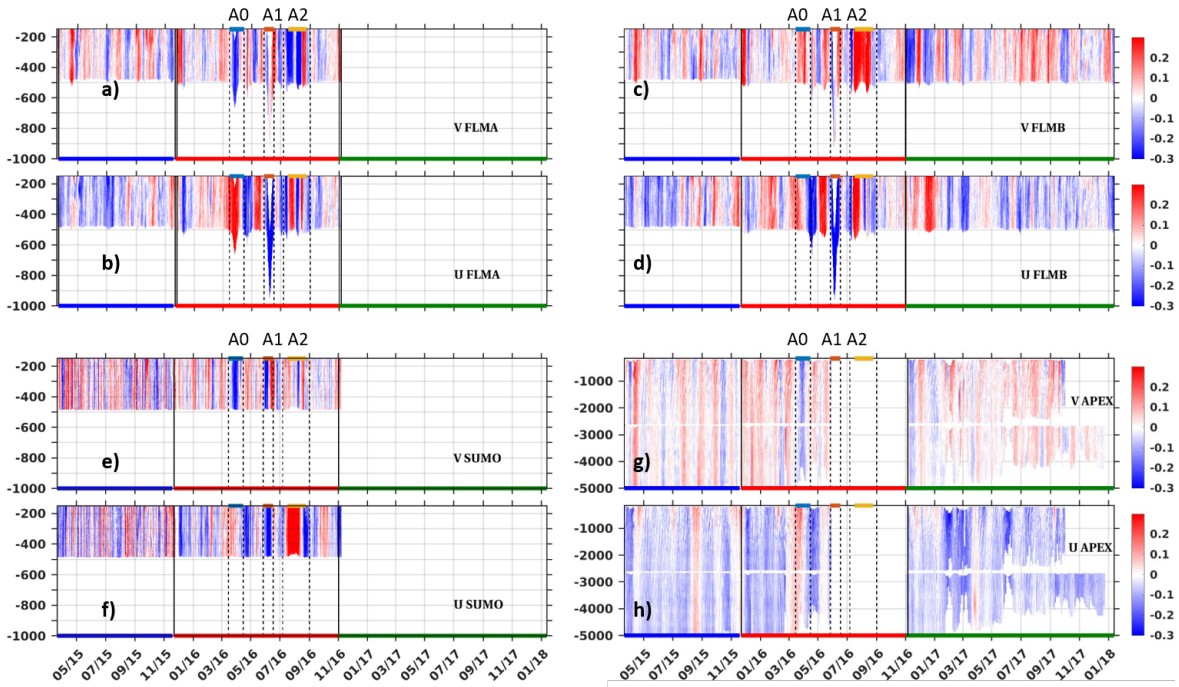

**Figure 5.** Meridional (V) and zonal (U) velocity time series (m/s) from FLMA ADCP (a and b), FLMB B ADCP (c and d), SUMO ADCP (e and f) and APEX Aquadopp (g and h). The upper level is 150 m at all moorings. Time sampling for the ADCPs was one hour while the APEX profiler made a velocity profile per day. The APEX profiler did not sample the first 250 m.

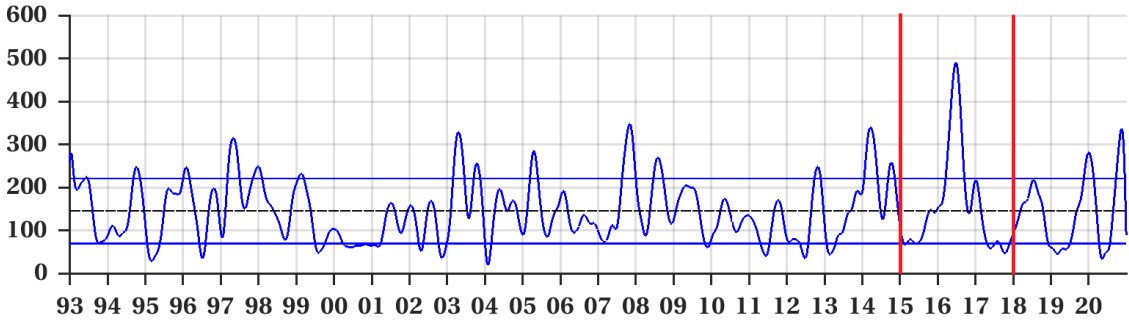

**Figure 6.** 180-day low-passed filtered altimetry derived EKE at the mooring array location. The horizontal black dashed line indicates the mean, the blue lines mark the value of the mean plus 1.5 standard deviation. The vertical red lines show the period of the mooring deployment.





A 29-year-long surface EKE time series was derived from satellite altimetry at the array site (Figure 6). The 29 year-long EKE time series does not feature any significant trend. Year 2016 stood out with EKE values of 400 x $10^{-4}$ m$^2$/s$^2$, that is about 3 times the mean value (145 x $10^{-4}$ m$^2$/s$^2$) (Figure 6).

## 3 The three events A0, A1 and A2

The three events featured daily average velocity amplitudes in excess of 0.3 m/s at the three moorings (SUMO, FLMA and FLMB)(Figure 7 a, b, c). The velocity signal at A0 was smaller (> 0.3 m/s) than at A1 and A2 (> 0.5 m/s). APEX mooring documented the A0 event and showed that A0 velocity anomalies were bottom reaching (Figure 5 g and h). During the A0 event (April 2016), vertically averaged meridional velocities at FLMB were of opposite sign to those at FLMA and SUMO while zonal velocities varied together (larger amplitude at FLMA) (Figure 7 d and e). During A1, velocities at the three moorings

were in phase (Figure 7 d and e). Event A2 comprised two velocity maxima at the 3 moorings, A2' (peaking at 0.7 m/s at FLMB) and A2" (0.8 m/s at SUMO) (Figure 7 a, b, c).

### 3.1 Identification of A0, A1 and A2 events in satellite altimetry

We compared altimetry-derived surface geostrophic velocities (SGV) time series at the 3 moorings to the vertically averaged in situ velocities time series during A0, A1 and A2. The vertical averages (full lines in Figures 7d and e) cover the depth

range 150-450 m for SUMO and variable depth ranges for FLMA and FLMB as the moorings underwent significant vertical excursions (Figures 7 a and b). The SGV times series (dashed lines in Figures 7 d and e) match vertically-averaged observations in terms of amplitude and direction (Figures 7 d and e). The agreement between SGV and in situ velocities is good during event A1 (root mean squared difference, rmsd, less than 0.08 m/s for u and v at all moorings). During events A0 and A2, satellite-derived meridional velocities at FLMA and FLMB are smaller than vertically averaged in situ velocities by about 0.2 m/s (50%

of the amplitude) and the rmsd (bias removed) are of the order of 0.1 and 0.09 m/s respectively (Figure 7 d).

We identified the structures associated with A0, A1 and A2 in the satellite altimetry maps (Figure 8). A video supplement shows the evolution of the SSH and velocity fields from January to September 2016 and provides information regarding the origin and fate of the structures.

A0 corresponds to the boundary of an elongated meander of the STF passing in the vicinity of the mooring array (Figure 8a),

A1 to a large anticyclonic eddy (Figure 8b) and A2 to two small anticyclonic eddies A2' and A2" (video supplement; Figure 8c shows A2").

The V-shape curvature of the A0 meander boundary near the array on March 30 explains the negative vertically averaged meridional velocities observed at FLMA and SUMO and the positive ones observed at FLMB (Figure 7d, 8a). FLMA, the closest mooring to the meander, experienced the largest velocities (0.45 m/s) and dived by 100 m (Appendix A1). The meander

had a spatial scale of the order of 600 km and consisted of three anticyclones. The anticyclone in proximity to the mooring array was shed from the STF at 36°W and 40°S on January 27 (blue cross in Figure 8d). It then re-joined the STF and contributed to the meander (dashed line in Figure 8d marks its trajectory). The meander rapidly propagated from 36°W-40°S to the mooring



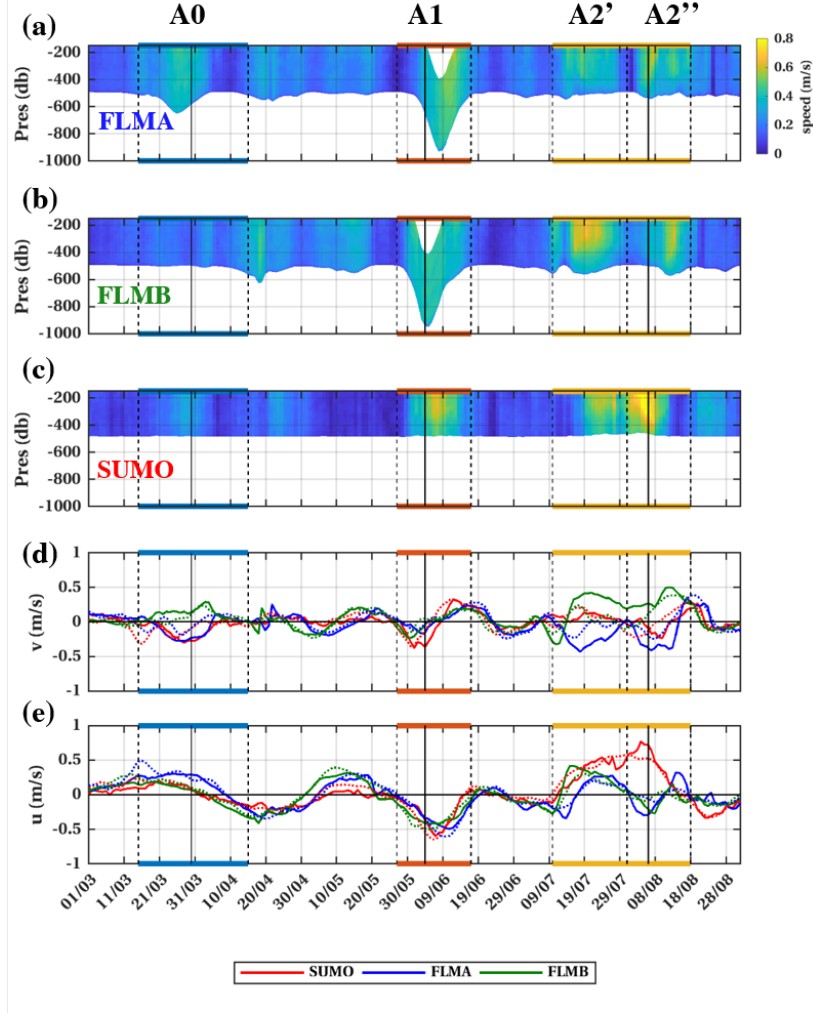

**Figure 7.** Daily averaged time series of velocity amplitude (m/s) from FLMA (a), FLMB (b) and SUMO (c). d-e) vertically averaged velocity time series from SUMO (red), FLMA (blue) and FLMB (green) : d) meridional component, e) zonal component. The dashed lines in d-e) are surface geostrophic velocities derived from satellite altimetry co-localised at the mooring locations. X-axis is time (mm/yy). Vertical dashed lines bound A0, A1 and A2' and A2". Black vertical lines correspond to dates considered in Figure 8

location in 10 days (advection velocity of 0.35 m/s). The meander broke apart on April 5 and the eddy was absorbed again in the STF on April 19 (at the location marked with the blue star in Figure 8d).

The large anticyclonic A1 eddy has a radius of about 150 km (Figure 8 b). The moorings sampled the northern part of the eddy (A1) on June 4 and the eastern edge on June 15 (video supplement). The westward displacement of the eddy between the two dates explains the change in sign (from negative to positive) observed in the in situ meridional velocity components (Figure 7). The eddy is advected with a zonal velocity of 0.15-0.2 m/s when passing through the mooring array while the maximum





**Figure 8.** a-c) SSH (m) and geostrophic velocities (arrows) from satellite altimetry for March 30 2016, June 4 2016 and August 6 2016. The vertically integrated in situ velocities are indicated with white arrows. d-f) SSH over the Argentine Basin at the same dates. Black isolines mark SSH every 0.1 m. The thick black and magenta contour correspond to the STF (SSH = 0.4 m) and the SAF (SSH = 0.05 m). The crosses indicate the region of formation of A0, A1, A2' and A2" at $time = t_0$. The pathway of each structure from its origin to the mooring site (pathway beyond the mooring site) is schematically indicated with black dashed (dotted) curves. The locations where the mesoscale structures are absorbed in the STF at time=$t_f$ are indicated with stars.

swirl velocity (estimated from the meridional velocity) is about 0.4 m/s (video supplement and Figure 8 b). A1 was shed from the STF at 40°S and 36°W (same location as A0 genesis) on March 29 (red cross in Figure 8e, animation in video supplement). A1 propagated westward with a speed of about 0.2 m/s reaching the mooring array by the beginning of June (Figure 8e). A1 was absorbed in the STF at the location of the red star in Figure 8e on June 30.




In contrast, satellite altimetry suggests that the two anticyclonic eddies contained in event A2 were small with a radii of 20 to 40 km and 50 km, respectively. The first eddy (A2') reached the mooring array on July 15 after a long journey. Indeed, A2' originated from the Brazil Current overshoot at 52°W 43°S on March 6 (yellow cross in Figure 8 f and video supplement), was entrained in the Zapiola Anticyclonic Circulation and reached the mooring array 4 months later. A2" detached from a meander of the STF at 40°W-42°S on July 18 (Figure 8 f) and reached the mooring array on August 4. Then it propagated northwestward and was absorbed by the STF on September 19 at the location of the yellow star in Figure 8 f.

A0 and A1 were generated in a region (40°S, 36°W) characterized by a strong gradient in planetary potential vorticity associated with a seamount (Figure 1a). A1 propagated southward and then westward following the 0.1900 10⁻⁷ f/H contour (Figure 8e, Figure 1b). A2' was advected from the Brazil Current overshoot in the Zapiola Anticyclonic Circulation along the 0.1925 10⁻⁷ f/H contour (Figure 8f, Figure 1b).

### 3.2 Hydrography of the extreme events: A0, A1 and A2

The temperature, salinity and density time series from the discrete unevenly distributed measurement levels were hourly averaged and vertically interpolated (Figure 9). Event A0, the "meander-like" structure (Figure 8 a and d), is best documented by FLMB as SUMO was at its periphery and FLMA dived by 100 m. The large A1 eddy (Figure 8 b) is best observed at SUMO as FLMA and FLMB underwent large draw-downs ($>$ 400 m) (Figure A1, Table 3). In contrast, A2 event hydrography is rather well documented at the 3 moorings.

A0 event at FLMB (March 15 to April 10) is associated with warm and salty waters (T $>$ 8°C and S $>$ 34.15 psu in the upper 200 m) and depressed isopycnals (down 100 m for isopycnal 27.0 kg/m³ reaching 400 m) (Figure 9 a, b, c). The warmest temperatures (17°C) recorded at SUMO were summer features and confined in the upper 30 m which were not sampled by FLMB (Figure 9 d).

In contrast, A1 and A2 events occurred during winter and showed homogeneous waters in the upper layer, which are likely to be associated with winter convection and deep mixed layers (down to 200 m for A1 in June, 350 m for A2' in July) (Figure 9 d-l). A1 featured temperatures larger than 8°C and salinities larger than 34.15 psu in the upper 400 m at SUMO (Figure 9 d and e). During A1 isopycnal 27.00 kg/m³ deepened down to 500 m. The vertical extension of A1 is at least 1500 m (Figure 3).

The hydrographic signals associated with A2' and A2" are larger at FLMA and FLMB than at SUMO (Figure 9 d-l; Figure 8 c, f). SUMO did not sample A2' and was on the southern edge of A2". A2' signatures are first observed at FLMB between July 9 and August 1 and then at the mooring A between July 15 and August 5 (Figure 9 g-l). The displacement of isopycnal 26.86 kg/m³ exceeded 300 m at FLMB and 200 m at FLMA. A2' eddy core came closer to FLMA than FLMB as FLMA recorded a thicker homogeneous layer, a larger isopycnal displacement and smaller velocities (Figure 9 g-l and Figure 7 a and b). A2" is first observed at FLMB, then at SUMO and lastly at FLMA with more homogeneous values at FLMB and smaller velocities indicating that the eddy center came closer to FLMB. A2' and A2" have vertical extensions of at least 1500 m (Figures 3 and A1). These large hydrographic vertical extents contrast with the rather small horizontal scales suggested by satellite altimetry.





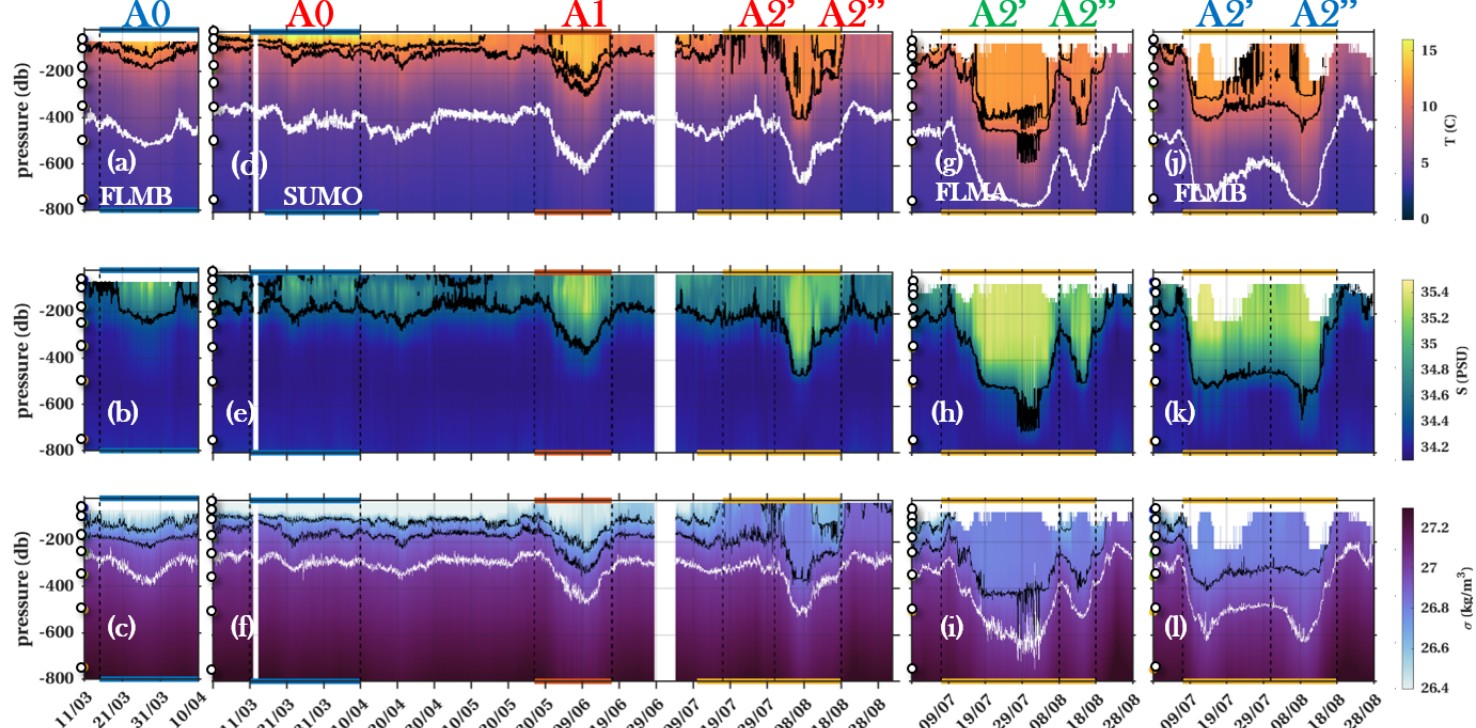

**Figure 9.** a-c) Temperature, salinity, density from mooring B (FLMB) during A0 event (hourly averaged data are linearly interpolated in the vertical). In (a) the black isolines are 10°C and 12°C isotherms and the white isoline is the 4.5°C isotherm. In (b) the black isolines is the 34.5 isohaline. In (c) the black isolines are the 26.65 and 26.85 kg/m³ isopycnals and the white isoline is the 27.00 kg/m³ isopycnal. d-f) Same as (a-c) at the surface mooring (SUMO) during events A0, A1 and A2. g-i) Same as (a-c) at mooring A (FLMA) during event A2 event. j-l) Same as (a-c) at mooring B (FLMB) during A2 event. White dots in the y-axis indicate the discrete levels used for the interpolation.

## 3.3 Eddy characteristics at the mooring array

The three vigorous anticyclonic eddies A1, A2' and A2" characteristics (SLA amplitude, radius, swirl velocity, life time, path length, propagation velocity, intensity, non linearity, vertical extension, relative vorticity, and Rossby number) are shown in Table 4. Eddy amplitude is the SLA difference between the eddy center and its periphery. Radius ranges were estimated from the 1/4° interpolated satellite gridded maps in which eddies had an elliptical shape. Therefore the uncertainty in the radius is reported as a range in Table 4. Intensity is the ratio of the eddy amplitude to its radius (Frenger et al., 2015) and non linearity the ratio of the eddy swirl speed to the propagation velocity (Chelton et al., 2011). Swirl velocities were estimated from in situ and satellite data. They differ by a factor of 2 for A2' which was attenuated in satellite altimetry maps. The order of magnitude





of the relative vorticity was estimated from in situ swirl velocities and satellite derived radius. The three anticyclones show distinct characteristics.

| | Source | A1 | A2' | A2" |
|---|---|---|---|---|
| amplitude A (cm) | satellite | 60 | 15 | 25 |
| radius R (km) | satellite | 100-150 | 20-40 | 40-50 |
| intensity A/R (cm/km) | satellite | 0.4-0.6 | 0.4-0.75 | 0.5-0.6 |
| life time (months) | satellite | 3 | 6 | 2 |
| path length (km) | satellite | 1200 | 2000 | 300 |
| swirl velocity V (m/s) | in situ | 0.4 | 0.4 | 0.5 |
| | satellite | 0.4 | 0.2 | 0.4 |
| propagation velocity c (m/s) | satellite | 0.2 | 0.05 | 0.1 |
| eddy non-linearity V/c | in situ | 2 | 8 | 5 |
| | satellite | 2 | 4 | 4 |
| relative vorticity $\zeta (1/s)$ | in situ & satellite | $2.6 -4 \times 10^{-6}$ | $1 - 2 \times 10^{-5}$ | $1 \times 10^{-5}$ |
| vertical extension (m) | in situ & satellite | 5000 | 2500 | 2500 |
| Rossby Number | in situ & satellite | 0.02-0.06 | 0.1-0.3 | 0.1-0.2 |

**Table 4.** A1, A2' and A2" properties when passing through OOI array.

A1 was a large anticyclone (SLA amplitude of 60 cm, radius of 100-150 km and Rossby Number less than 0.1). Event A2 comprised two smaller anticyclones A2' and A2" (amplitude of 15 and 25 cm and radius of 20-40 and 40-50 km and 0.1<Rossby Number<1, respectively). The three eddies A1, A2' and A2" were intense (intensities of 0.4 - 0.5 cm/km), with similar swirl velocities (about 0.4 -0.5 m/s) and were non-linear advecting trapped fluid thus transporting water properties (Table 4). A2' and A2" underwent winter convection with deep mixed layers in excess of 350 m at FLMA and FLMB.

To estimate the vertical extent of the eddies, we compared time series of dynamic height anomaly at the mooring sites estimated from the in situ data to the satellite data (Appendix A3). This comparison suggests that A1 was a bottom reaching structure while A2' and A2" extended down to 2500 m at the three moorings.

We now analyze the high frequency variations associated with the mesoscale structures.

## 4 High frequencies during extreme events

### 4.1 Velocity spectral content at high frequencies

Rotary spectra of the ADCP velocities were produced considering 30-day long time series spanning each event, when the moorings did not experience vertical motions. SUMO data were used for the three events (A0, A1, A2") and FLMB data





for A0 (Figure 10). A2' rotary spectra could not be built as A2' event was not sampled at SUMO and induced large vertical displacements in FLMA and FLMB.

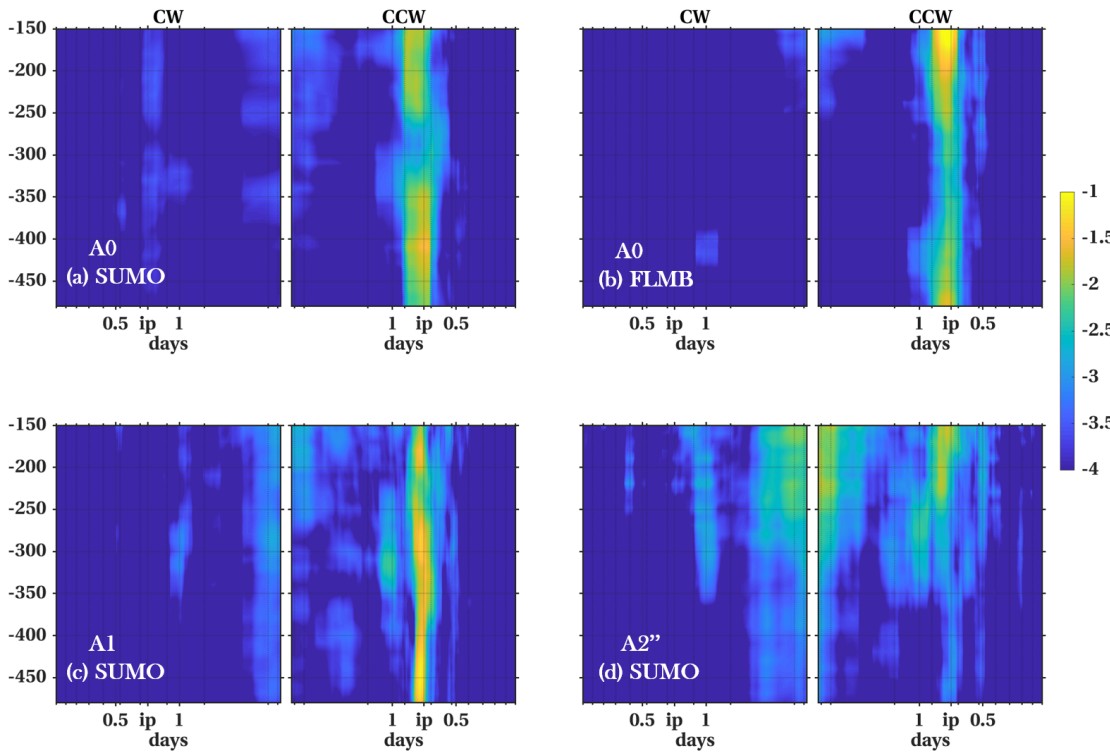

**Figure 10.** Velocity rotary spectra for A0 event at surface mooring (SUMO) (a), for A0 event at mooring B (FLMB) (b), for A1 event at the surface mooring (c) and A2" at surface mooring (d). The x axis is periods in days. IP is the inertial period (17 h at the mooring latitude). The y axis is depth in meters. Colorbar is log of energy in $m^2/s^2$

All the rotary spectra show a clear peak in energy (of about 0.1 $m^2/s^2$ ) near the inertial frequency (period of 17 h at the mooring location) in the counterclockwise motions over the ADCP depth range (down to 480 m). The peaks show a red shift (larger period) relative to the inertial period. The spectra show little ($< 0.01$ $m^2/s^2$) or no energy at the diurnal or semi-diurnal period (Figure 10). The velocity spectra thus indicate significant motions associated with near-inertial waves (NIW) and negligible tidal currents apart from a low energy signal at middle depth (200 - 350 m) at the diurnal period at A1 and A2

which could be associated with internal tidal waves at the pycnocline level (Figure 9 f).

Event A0 is documented with two spectra (Figure 10 a and b) from SUMO and FLMB ADCP data, as FLMA dived by 100 m (Figure A1). The peak near the inertial frequency in counterclockwise motions goes through a local minimum around 300-350 m in both spectra. The maximum observed above 200 m in the near inertial frequency band could be associated with locally generated NIW forced by winds whereas the local maximum below 350 m may be associated with trapped NIW (e.g.



Martinez-Marrero et al., 2019, Kawaguchi et al., 2020). The energy maximum at 450 m at a slightly larger period than the inertial period during A1 is another possible indication of trapped waves (Figure 10 c).

Indeed, the horizontal velocity shear of anticyclonic eddies $\zeta$ shifts f to a lower "effective planetary vorticity" ($f_{eff} = f + \zeta/2$) (Kunze, 1985). In the southern hemisphere (f<0), positive $\zeta$ of anticyclonic eddies result in $|f_{eff}| < |f|$ in their cores. When NIWs are exited inside a region of anticyclonic relative vorticity, they can have frequencies below f (red shift) and thus remain

trapped as they cannot propagate out of the rotating region. This results in an accumulation of wave energy in a critical layer following tilted isopycnals, and, eventually part of the energy is dissipated by buoyancy release through vertical mixing (e.g. Kunze, 1985; Kunze et al., 1995; Martinez-Marrero et al., 2017; Kawaguchi et al., 2020). Using the $\zeta$ values reported in Table 4, $f_{eff}$ approximately corresponds to a period of about 18 h, consistent with the red shifts observed in Figure 10 spectra.

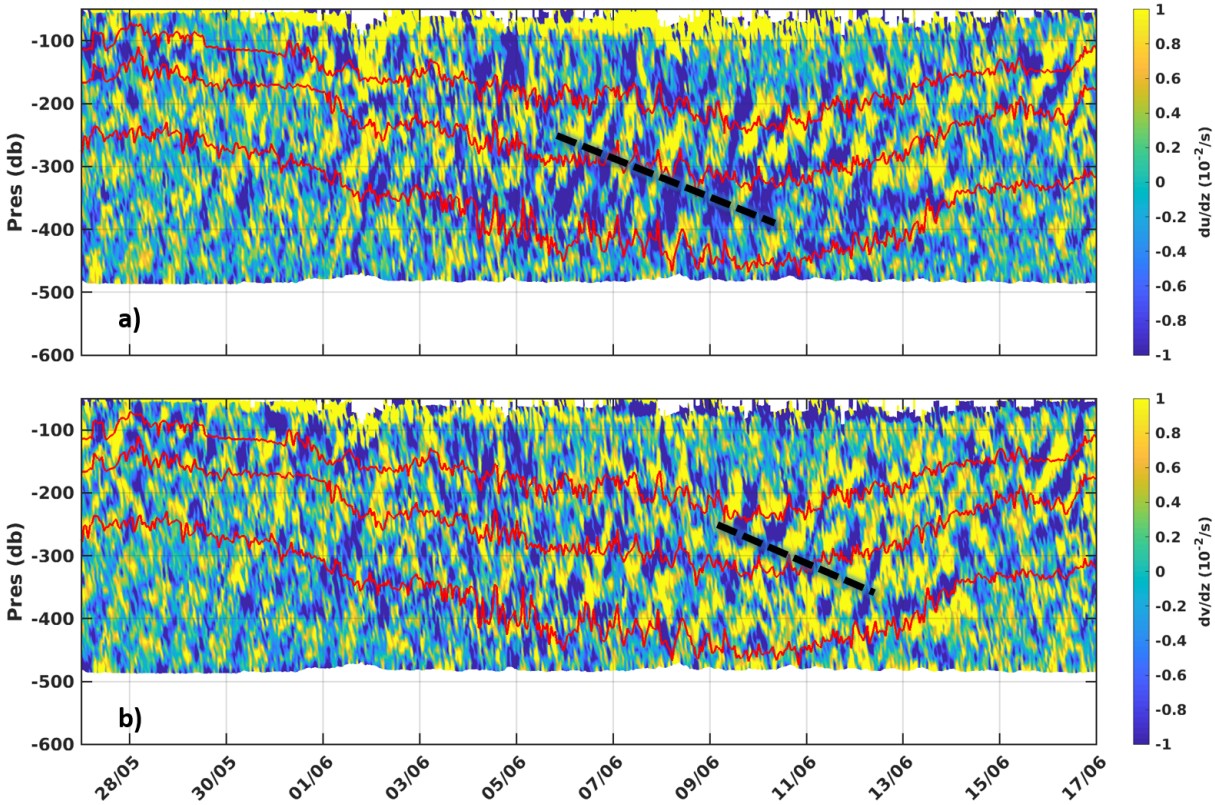

**Figure 11.** Vertical shear of velocity components for SUMO during event A1. Red contours correspond to isopycnals 26.65, 26.85 and 27 kg/m$^3$ computed from the vertically interpolated data. The black dashed lines indicate approximately the downward energy propagation.

As an example of the wave activity we show the vertical shear of the horizontal velocity components and density contours

for A1 (Figure 11). The vertical shear features clear wavy patterns close to the inertial period with downward group velocity of the order of 30 m/d (black dashed lines in Figure 11). The vertical wavelength is about 50 m (Figure 11 a and b). The pattern is





intensified at depth following isopycnals 26.65 and 27.00 kg/m$^3$ between June 4 and 12 (Figure 11 a and b). We now examine the occurrence and amplitude of NIW signals in density and velocity shear fields for all the events.

## 4.2 Near-inertial waves in the density and velocity shear field

We applied a third-order Butterworth band-pass filter around the inertial frequency (14h-20h) to the shear and density time series to isolate the signal associated with the near-inertial waves and computed the envelopes of the filtered signals. Envelopes of the band-pass filter time series measure the amplitude of the fluctuations associated with NIW. The filtered density times series at 180 m for events A0, A1 and A2 at SUMO are shown in Appendix A4 as an example.

During A0, A1 and A2" the shear envelopes show large values at the surface down to 400 m (Figure 12 a). The NIWs could
be locally generated by intense winds that characterize the region (red colours from Figure 12 b). The large shear envelopes at depth occur at times of positive satellite altimetry-derived $\zeta/f$ (Figure 12 c) confirming that NIWs are trapped in the anticyclonic structures. Note that the temperature at 350 m shows high-frequency variations during the three events (Figure 12 c).

The envelopes of the NIW density signature at each measured level in the upper 500 m at SUMO are shown in Figure 12
d. Large envelope values ($> 0.02$ kg/m$^3$) are observed in the upper 150 m during summer when waters are stratified. Winter envelope maxima occurred near the surface when isopycnals are almost vertical. Local maxima below 150 m extend quite deep in the water column during the three anticyclonic events between isopycnals 26.65 and 27.00 kg/m$^3$ (Figure 12d).

The energy from the trapped waves must then be transferred to smaller waves, which eventually break generating turbulence and mixing (Kunze et al., 1995). The tendency of a stratified water column to become unstable can be estimated with the
Richardson number $R_i= N^2/S^2$ where $N^2$ is the Brunt-Väisälä frequency and $S^2$ the squared velocity shear ($S^2= (\partial u/\partial z)^2$ + $(\partial v/\partial z)^2$). The Richardson number is often used as a proxy to predict the likelihood of overturning events and enhanced mixing in a stratified fluid, due to shear instability acting against the stable buoyancy field. Though the existence of a universal critical Richardson number $Ri_c$ at which turbulent mixing can start is still debated, a lower bound Ri $< Ri_c$ is often assumed, for example $Ri_c$ = 1/4, originally derived from linear stability of steady stratified shear flows (Miles, 1961), or $Ri_c$ = 1 from
non-linear stability analysis (e.g Abarbanel et al., 1984).

## 4.3 Richardson number

The Brunt-Väisälä frequency (Figure 13 first row) was estimated from interpolated densities shown in Figure 9 (only 7 to 8 measurement levels). The Richardson number (Figure 13, last row) was tentatively computed as the daily minimum of the ratio between $N^2$ and the squared amplitude of the vertical shear of the horizontal velocity (Figure 13, second row).

A0 event (March 15 to April 10), best appreciated at FLMB (Figure 13 a, b, c) with a larger depression of the 27.00 isopycnal than at SUMO, is associated with rather large $N^2$ values compared to surrounding waters (Figure 13 a). The velocity shear amplitude features large values in the upper stratified 150 m and, interestingly, a local maximum below 27.00 kg/m$^3$ isopycnal at its deepest level (400 m) resulting in a rather small Richardson number (0.75) (Figure 13 b and c).



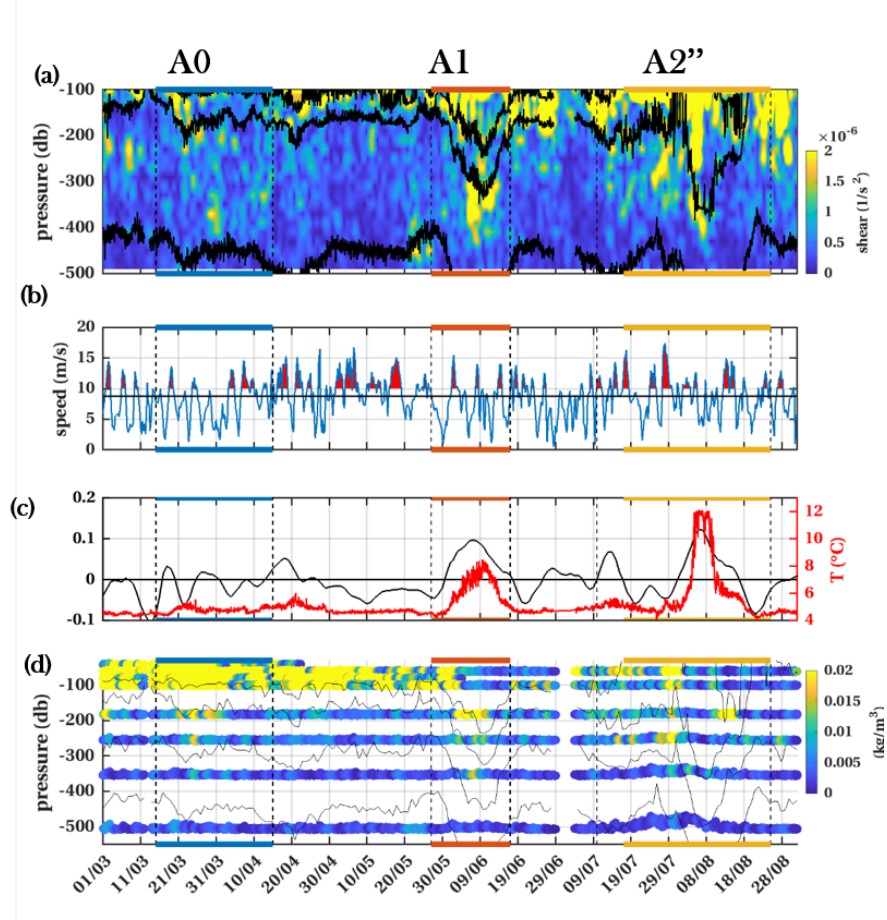

**Figure 12.** a) Envelope of the 14-20 h band-pass filtered shear intensity at SUMO. Black isolines are 3 isopycnals computed from SUMO vertically interpolated density time series: 26.65, 26.85, 27.00 kg/m$^3$. b) Wind intensity from Era-Interim at SUMO. Red colors indicate values larger than 10 m/s. c) Satellite altimetry derived $\zeta/f$ (Left y-axis) and temperature at 350 m at SUMO (Right y-axis) d) Envelope of the 14-20 h band-pass filtered density at SUMO.

A1 is sampled in austral fall (end of May beginning of June) and features low $N^2$ (less than 0.1 x 10$^{-5}$ s$^{-2}$ in the upper 200 m) on June 9 when isopycnals reach their maximum depth (Figure 13 d). Below this homogeneous upper core, SUMO documented stratified waters with large $N^2$ values ($N^2 > 2.5$ x 10$^{-5}$ s$^{-2}$), a local maximum in the shear intensity around isopycnal 27.00 kg/m$^3$ (when deeper than 300 m) (4 x 10$^{-5}$ s$^{-2}$), that led to a Richardson number of about 0.62 (Figure 13 d, e, f). A strickingly low Richardson number (Ri = 0.32 from May 20 to 30) is observed at depth (below 400 m) just before A1 event and corresponds to a vigorous velocity vertical shear at a time of steeply shallowing isopycnals (Figure 13f).

Anticyclones A2' and A2" have a deep reaching (350 m at FLMB and 250 m at FLMA) homogeneous layer of low $N^2$ ($N^2 < 0.5$ x 10$^{-5}$ s$^{-2}$) corresponding to deep winter mixed layers (Figure 13, g and j). Below the homogeneous mixed layer,



a local maximum of $N^2$ of the order of 1.5 x 10$^{-5}$ s$^{-2}$ is centered around the 26.85 kg/m$^3$ isopycnal. The velocity shear $S^2$ is small in the mixed layer and increases below the 27.0 kg/m$^3$ isopycnal (Figure 13 h and k). The Richardson number shows two minima in the vertical: one in the deep winter mixed layer dominated by extremely small $N^2$ and another one below the 27.0 kg/m$^3$ isopycnal where velocity shear is important (Ri of about 0.6 to 0.7) (Figure 13 g,h,i,j,k and l). The two minima are separated by a layer of large Ri (3.16) which corresponds to $N^2$ values of 1.5 x 10$^{-5}$ s$^{-2}$.

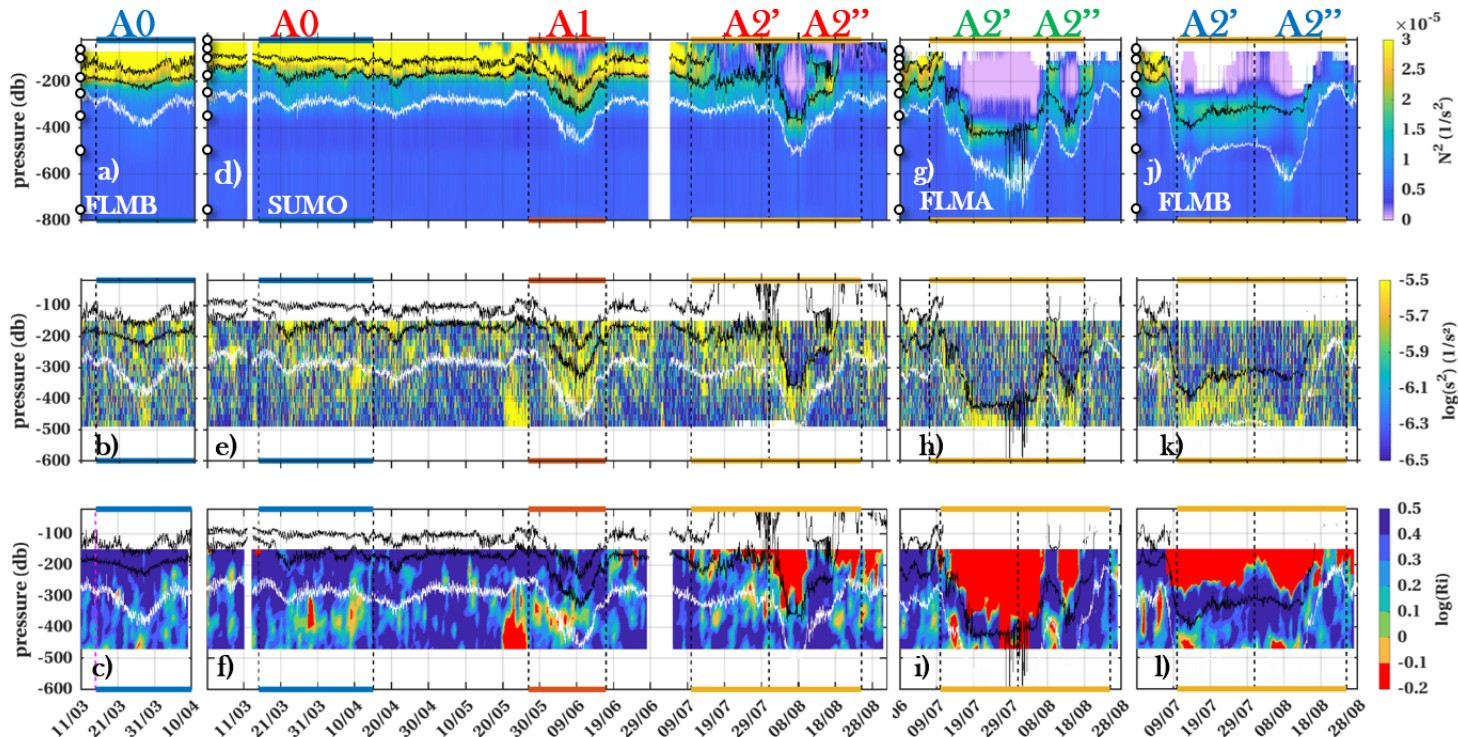

**Figure 13.** a-c) Brunt-Väisälä frequency ($N^2$) (hourly averaged data are linearly interpolated in the vertical), amplitude of vertical shear and Richardson Number during event A0 at mooring B. The Richardson number was tentatively computed as the daily minimum of the ratio between $N^2$ and the vertical shear of the horizontal velocity. The black isolines are the 26.65 and 26.85 kg/m$^3$ isopycnals and the white isoline is the 27.00 kg/m$^3$ isopycnal which corresponds to the limit between SACW and AAIW. White dots in the y-axis in (a) indicate the discrete levels used for the interpolation.

d-f) Same as (a-c) at the surface mooring during events A0, A1 and A2. g-i) Same as (a-c) at mooring A during event A2 event. j-l) Same as (a-c) at mooring B during A2 event.

The four anticyclonic features show a low Ri at depth (between 300 and 400 m for A0 and A1 and between 400 and 500 m for A2' and A2") at the limit between SACW and AAIW (indicated with a white isoline in Figure 13). The low Ri suggests that favourable conditions for enhanced mixing between AAIW and SACW occurred at depth within the anticyclones where NIWs are trapped and eventually break.



## 5 Conclusions

We analysed physical parameters recorded by the OOI mooring array deployed at 42.5°S and 42.5°W a remote site in the data-sparse Argentine Basin. The moorings were located 3° to the south of the Subtropical Front mean position and to the north of the Zapiola Anticyclone in a region characterized by rough weather conditions and relatively low EKE (145 x $10^{-4}$ m$^2$/s$^2$) compared to the surroundings (EKE can be as large as 1000 x $10^{-4}$ m$^2$/s$^2$ in the Brazil Malvinas Confluence) (Figure 1).

Year 2016 stands out with a maximum of EKE (> 400 x $10^{-4}$ m$^2$/s$^2$) in the 29-year-long satellite altimetry record at the mooring site. We investigated possible circumstances (ocean circulation, wind anomalies) leading to this unique EKE peak in 2016 without any conclusive results. This needs further investigation. The mooring array documented four exceptional events in 2016. We tentatively address the questions raised in the introduction.

What are these extreme events? The four events were anticyclonic structures, a meander of the Subtropical Front (A0) and three intense anticyclonic eddies with swirl velocities of the order of 0.4 m/s. Satellite altimetry provided information on the size, origin and fate of the structures. The three anticyclonic eddies (named A1, A2' and A2") showed distinct characteristics. A1 was a large, 200-300 km-diameter, bottom-reaching eddy (bottom > 5000 m). In contrast A2' and A2", were smaller and attenuated in the satellite altimetry maps. They had radii of about 40 km (close to the Rossby radius, with A2' probably smaller and A2" slightly larger) and a vertical extent of about 2500 m.

Where do those anticyclonic structures come from? Satellite altimetry suggested that the three eddies had different origins and paths. A1 was generated from an STF instability 600 km to the northeast of the array over a topographic rise (40°S, 36°W) 60 days before crossing the OOI array. A2' originated from the Brazil Current overshoot (43°S, 52°W) as large and strong eddy and travelled during 4 months with the Zapiola Anticyclonic Circulation before reaching the array. While at the array, satellite altimetry maps suggest that A2' merged with A2" which arrived after a 16-day trip from its birth place at 42°S, 40°W. After a time life of 3, 6 and 2 months for A1, A2' and A2" respectively, the three eddies were reabsorbed by the STF.

How often do the extreme events occur? The altimetry-derived EKE time series at the mooring location showed that those events were exceptional (Figure 6). In contrast, eddies detached from the subtropical front are frequent in the energetic Argentine Basin (Figure 1), in particular in the Brazil-Malvinas Confluence, in the vicinity of the STF and in a band centered at 49°S (e.g. Fu, 2006; Mason et al., 2017; Artana et al., 2019). Thus, OOI observations in 2016 are relevant and valuable for the data sparse Argentine Basin.

Do the anticyclonic structures documented with OOI impact mixing? These anticyclonic structures are not subtropical "run-away" eddies lost to the subantarctic waters, they were reabsorbed in the subtropical waters. The mooring data provided evidence of favourable conditions for the potential development of mixing well below the mixed layer. Indeed, the estimated Richardson Number showed two minima in the vertical during the extreme events, one in the mixed layer and another one below the mixed layer depth at the 27.00 kg/m$^3$ horizon associated to NIWs suggesting mixing between the SACW and the AAIW. Moreover, temperature at 350 m shows large variations at high frequencies (< 5 h) during A1 and A2 suggesting that energy cascading may be occurring (not shown). Regardless of their size and origin, the four anticyclonic structures presented



low Ri associated with trapped NIWs suggesting that enhanced mixing at depth is probably found in the many anticyclonic eddies populating the Argentine Basin. Those numerous anticyclonic mesoscale eddies likely act as mixing structures at the pycnocline and bring heat and salt from the SACW to the AAIW and they are certainly relevant in modifying the upper water masses characteristics in the Argentine Basin.

The OOI array provided unprecedented long time series with a high temporal sampling in the Southern Hemisphere and,
for the first time documented near-inertial waves trapped at depth within anticyclones in the Argentine Basin. The majority of observational evidence of NIWs trapping in anticyclonic structure are from the Northern Hemisphere and rely upon quasi-synoptic continuous vertical density profiles allowing estimates of NIWs vertical scales, small structures in the Brunt-Väisälä frequency and dissipation rates (e.g. Joyce et al., 2013; Karstensen et al., 2017; Martinez-Marrero et al., 2019) and recently upon mooring data (e.g. Kawaguchi et al., 2020 and 2021; Xu et al., 2022; Yu et al., 2022 and Ma et al., 2022).

Precise examination of NIWs is beyond the scope of this work and deserves further investigation. Recent analyses of mooring observations in the Northern Hemisphere have provided new insights on NIWs trapped in anticyclonic eddies. Kawaguchi et al. (2020) highlighted trapped amplified NIWs with multiple inertial frequencies (double, triple and quadruple) generated by a fast-moving cyclone (winds > 20 m/s) passing over an anticyclone in the central Sea of Japan. The OOI array did not detect multiple inertial frequencies in the anticyclones. In the northern South China Sea, Xu et al. (2022) evidenced an anticyclonic
eddy carrying typhoon generated trapped NIWs for at least 660 km and 79 days. The precise wind episodes leading to the NIW generation in the anticyclones observed at OOI were not clearly identified. ERA-Interim winds showed various strong wind events (wind speed larger than 20 m/s) along the eddies paths. NIW penetration down to 1400 m close to the seabed was observed in the northwestern South China Sea (Ma et al., 2022). The vertical range of the OOI ADCP data prevented examining the presence of NIW below 500 m and the vertical extent of the NIW penetration is an open question as A1 was a
bottom reaching eddy and A2' and A2" had a vertical extension of 2500 m. Kawaguchi et al., (2021) documented amplified signals of NIW-related vertical shear and turbulent kinetic energy dissipation between cores of pair vortices (adjacent cyclone and anticyclone). A striking low Ri was observed at depth below 400 m just before A1 event at SUMO. The low Ri values are found between eddy cores of a tripole structure crossing the mooring array (not shown). Further investigation is needed to prove the link between the tripole and the low Ri values. Mooring observations at a submesoscale sampling in the northeast
Atlantic, a region with moderate EKE, suggested that submesoscale motions with Rossby Number close to 1 are limited to the mixed layer and have little effect on the trapping and vertical penetration of NIWs (Yu et al., 2022). The OOI eddies have Rossby numbers between 0.01 and 0.6, and A2', blurred in the satellite altimetry maps, could be considered as a submesoscale feature. The interactions between submesoscale features and NIWs could be different in a highly energetic region like the Argentine Basin where submesoscale motions are possibly not limited to the mixed layer.

*Data availability.* The mooring data, available at https://oceanobservatories.org/array/global-argentine-array/, is based upon work supported by the National Science Foundation under Cooperative Agreement No. 1743430 which supports the Ocean Observatory Initiative. The



altimeter products were produced by Developing Use of Altimetry for Climate Studies (DUACS) (http://www.aviso.altimetry.fr/duacs/) and distributed by Copernicus Marine Service (CMEMS) (http://marine.copernicus.eu/).

*Video supplement.* The animation (https://doi.org/10.5446/60932) shows the evolution of the SSH and the geostrophic velocity fields (in arrows) from satellite altimetry from January to September 2016. The vertically integrated in situ velocities (upper 400 m) are indicated with cyan arrows. The SSH data set is an up-to-date delayed-time daily product with a 1/4° ×1/4° spatial resolution, generated by the Developing Use of Altimetry for Climate Studies (DUACS) system and distributed by Copernicus Marine Service (CMEMS, http://marine.copernicus.eu/.). It merges the observations of several satellites, up to five at a given time (Jason-2, Jason-3, cryosat, haiyang, altika, and sentinel3a)

## Appendix A

### A1    Hydrography at FLMA, FLMB and APEX

FLMA and FLMB did not move much during deployments 1 and 3 while they underwent large draw-downs during deployment 2 ( Figure A1). A0 event led to vertical excursions of 100 m at 1500 m depth at FLMA while FLMB remained still (Figure A1). As a result, A0 event signed with temperature and salinity maxima in FLMB time series and minima in the time series in the upper layer of the blown-down FLMA (Figure A1).



**Figure A1.** Temperature (a,d,g), salinity (b,e,h), and pressure (c and f) time series at different depths (indicated with colors) from mooring A, B and APEX profiler mooring instruments respectively. Three events are labelled: A0, A1 and A2.

Event A1 led to vertical excursions in excess of 400 m at FLMA and FLMB and to the collapse of the APEX profiler in May 2016 (Figure A1 c, f, g and h). As a result of the draw-downs, the temperature (salinity) time series at FLMA and FLMB feature a local minimum in the upper 1000 m (250 m) during A1 instead of a maximum as in SUMO (Figure A1 and Figure 3). In contrast, FLMA and FLMB did not dive during A2 which signs as local maxima in temperature and salinity (down to 1500 m in temperature and 500 m in salinity) (Figure A1).






The APEX profiler data with a continuous vertical resolution of about 2 m between 250 to 1500 m clearly show the limits of the Antarctic Intermediate Water between 250 and 750 m (salinity minimum) and Upper Circumpolar Deep Water below (Figure A1). The APEX mooring did not sample A1 and A2 events due to its collapse.

## A2  Horizontal velocity statistics

| | | Deployment 1 | | | | Deployment 2 | | | | Deployment 3 | | | |
|---|---|---|---|---|---|---|---|---|---|---|---|---|---|
| | | u($\mu/\sigma$) | v($\mu/\sigma$) | V($\mu/\sigma$) | EKE | u($\mu/\sigma$) | v($\mu/\sigma$) | V($\mu/\sigma$) | EKE | u($\mu/\sigma$) | v($\mu/\sigma$) | V($\mu/\sigma$) | EKE |
| FLMA | hourly | -4/8 | 0/9 | 13/7 | 83 | -1/16 | -1/14 | 18/11 | 234 | - | - | - | - |
| | daily | -4/7 | 0/9 | 10/7 | 72 | -1/16 | -1/13 | 17/11 | 224 | - | - | - | - |
| | SGV | -1/8 | 0/8 | 11/5 | 76 | 1/17 | 1/11 | 18/10 | 226 | -3/9 | 1/10 | 12/7 | 100 |
| FLMB | hourly | -3/10 | 0/8 | 13/6 | 84 | -1/16 | 4/14 | 18/11 | 232 | -2/9 | 0/10 | 13/6 | 100 |
| | daily | -3/8 | 0/8 | 9/6 | 72 | -1/16 | -1/13 | 17/11 | 224 | -2/8 | 0/10 | 11/8 | 91 |
| | SGV | 0/10 | 0/9 | 12/6 | 96 | 3/17 | 3/11 | 18/10 | 219 | 0/10 | 1/11 | 13/7 | 113 |
| SUMO | hourly | -2/13 | 1/13 | 24/9 | 185 | -1/19 | 0/11 | 26/12 | 259 | - | - | - | - |
| | daily | -2/6 | 1/7 | 9/5 | 50 | -1/18 | 0/10 | 16/13 | 229 | - | - | - | - |
| | SGV | 0/9 | 1/9 | 11/6 | 86 | 1/19 | 2/11 | 18/14 | 265 | -5/9 | 2/8 | 12/7 | -87 |
| APEX | daily (250-500 m) | -3/5 | 1/7 | 9/4 | 41 | - | - | - | - | -6/8 | 2/7 | 11/6 | 63 |
| | daily (500-2500 m) | -3/5 | 0/5 | 7/3 | 27 | - | - | - | - | -7/5 | 1/5 | 9/5 | 28 |
| | daily (3000-4000m) | -5/4 | 1/4 | 8/2 | 21 | - | - | - | - | -6/4 | 1/4 | 8/3 | 21 |
| | SGV | 0/8 | 1/9 | 11/6 | 82 | 0/19 | 2/11 | 18/13 | 250 | -6/9 | 2/8 | 12/8 | 83 |

**Table A1.** Horizontal velocity statistics: mean ($\mu$), standard deviation ($\sigma$) (in $10^{-2}$ m/s) and EKE (in $10^{-4}$ m$^2$/s$^2$). In situ velocities were vertically averaged. Vertical averages cover the depth range 150-450 m for SUMO and variable depth ranges for FLMA and FLMB because of the vertical excursions. SGV stands for daily altimetry-derived surface geostrophic velocity.

## A3  Vertical extent of the eddies

To estimate the vertical extent of the eddies, we compared time series of dynamic height anomaly at the mooring sites estimated from the in situ data to the satellite data (Figure A2). As the mooring data suggest that the three eddies were structures reaching below 1500 m (section 3.2), we tentatively complemented the hydrographic mooring data below 1500 m with the CTD data of the deployment cruise. Various reference levels were considered in the estimation of the dynamic height anomaly.

Dynamic height anomaly computed using a 5000 m reference level well match (rmsd 9 cm) the along track SLA during event A1 at the SUMO location suggesting that A1 was a bottom reaching structure (Figure A2 b). In contrast, the best match between the different dynamic heights and the along-track SLA during A2' and A2" is found with a reference level of 2500 m at the three moorings (rmsd of 10 cm at SUMO, 15 cm at FLMA and FLMB) suggesting that both structures were shallower than A1 (Figure A2 b, c, d).



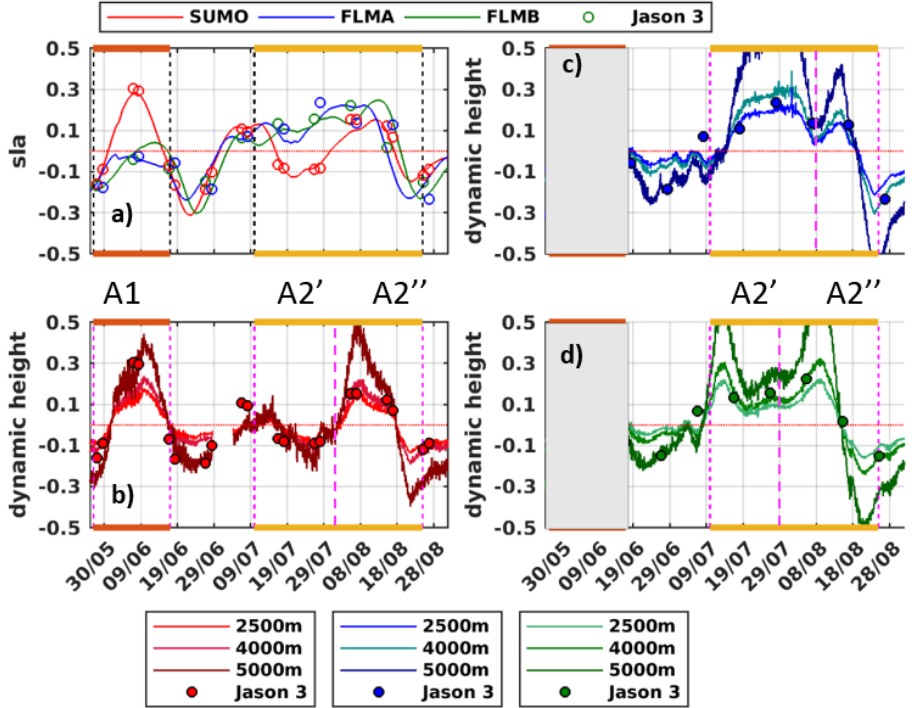

**Figure A2.** a) SLA time series from satellite altimetric maps (full lines) and Jason 3 satellite tracks (circles) interpolated at the mooring location (SUMO in red, FLMA in blue and FLMB in green). b) Dynamic height anomaly computed from vertically interpolated temperature and salinity referred to 2500, 4000 and 5000 m for SUMO and SLA from Jason at the mooring location. c) same as b) for FLMA. d) same as b) for FLMB. Shaded areas in c and d correspond to severe draw down during A1. The mean between the period May 26 to September first was removed to obtain the SLA and the dynamic height anomaly.

**A4 Density envelopes**

We applied a third-order Butterworth band-pass filter around the inertial frequency (14h-20h) to the density time series to isolate the density signal associated with the near-inertial waves and computed the envelopes of the filtered signals. The filtered density times series at 180 m for events A0, A1 and A2 at SUMO are shown as an example (Figure A3). Density was available at discrete unevenly distributed depths. Stratification was tentatively estimated with the Brunt-Väisälä frequency 390 ($N^2 = -g/\sigma \, \partial\sigma/\partial z$) calculated from the vertically interpolated temperature and salinity time series. Envelopes of the band-pass filter density time series measure the amplitude of the density fluctuations associated with NIW (red curves in Figure A3 a-c). The filtered density time series tend to show larger amplitude variations when stratification increases ($N^2$ above 1.5 x $10^{-5}$ $S^{-2}$) and smaller ones when stratification is weak. This is particularly stricking during A2" event: when $N^2$ is close to zero from August 4 to 10, the NIW signal in density reduces to less than 0.01 kg/m$^3$ and increases to 0.4 kg/m$^3$ when $N^2$ exceeds 3 $10^{-5}$ 395 from April 15 to 17 (Figure A3 c).

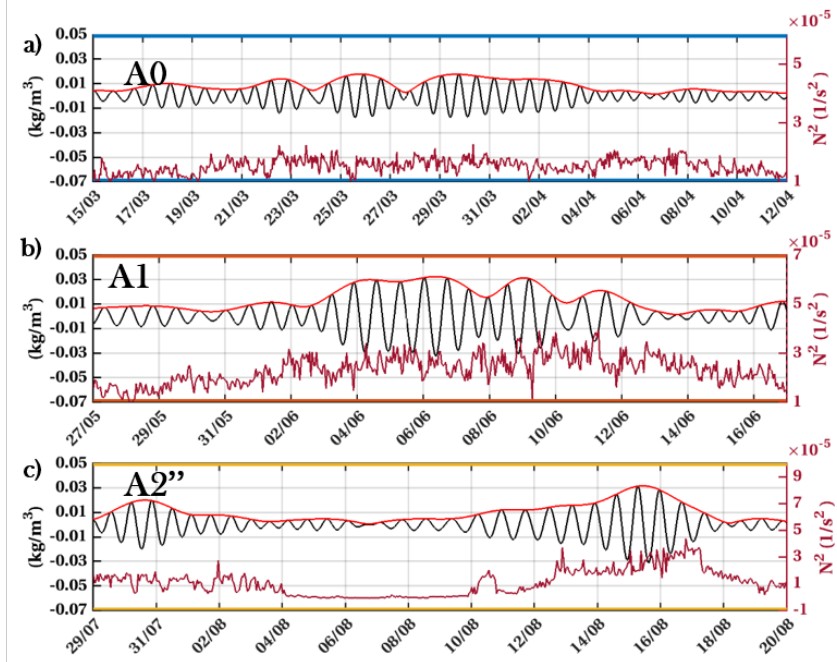

**Figure A3.** a-c) (14-20 h) band-pass filtered density time series (in kg/m$^3$) at 180 m (black), envelope of the filtered signal in red (left axis) and Brunt-Väisälä frequency time series at 180 m in brown (right axis) for event A0 (a), A1 (b) and A2 (c) from SUMO.

*Author contributions.* CA analyzed the data and CA and CP wrote the manuscript

*Competing interests.* The authors declare that they have no conflict of interest.

*Acknowledgements.* We are deeply grateful to scientists and technicians from Woods hole Oceanographic Institution and Scripps Institution of Oceanography who developed and operated the Argentine Basin array within the NSF funded Ocean Observatory Initiative. We are grateful to CNES (Centre National d'Etudes Spatiales) for constant support. This study is a contribution to the CNES-funded OSTST BACI project. Camila Artana acknowledges support from a CNES Postdoc Scholarship and funding from the Spanish government (AEI) through the 'Severo Ochoa Centre of Excellence' accreditation (CEX2019-000928-S).




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
