# Peer review of "Intense Anticyclones at the Global Argentine Basin Array of the Ocean Observatory Initiative"

_EGUsphere, 2023_

## Author Comment (AC1)

Summary and recommenndation: The authors analysed multi-year observations from a mooring array deployed in the Argentine Basin, and focued on a few intense anticyclonic eddy events during when near-inertial waves were also excited and mixing was likely to occur. The manuscript is overall clear and well written, and in my opinion, only minor revisions are needed so that the manuscript may be considered for publication.

Thank you for your comments and your time

General comments:

Figure 3: It seems that there are other extreme events occurred around 04/15, 12/16, 03/17, 04/17. Are these also events of mesoscale structures? Could you add some descriptions these events as well?

Indeed, in these dates, the time series show large anomalies. However, the salinity anomalies at 180 m did not exceed the threshold (mean + 3 std) that we established to consider the events as extreme. The SSH maps for those dates also show anticyclonic mesoscale structures. Below are snapshots of the SSH corresponding to the dates when eddies approached the moorings. These events are also observed in the new EKE time series (new Figure 6).

We added the following sentence in the conclusion in line 315:" Note that other anticyclonic eddies less extreme do occurred at OOI: they are seen as peaks in OOI hydrographic data and eke time series"

[Figure]

Figure 1: a-c) SSH (m) and geostrophic velocities (arrows) from satellite altimetry in April 8 2015, December 31 2016 and March 21 2017.

Figure 6: A large fraction of mesoscale signals would be filtered out when you used the a low-pass window of 180 days. Why don't you use a shorter temporal window like 30 days? Also, is there a mean flow in this region?
Thank you for your comment. We modified the figure using a low-pass window of 30 days (see figure below). We also observe that the year 2016 had the largest EKE values, which lasted for a long time. Indeed, there is a small mean flow. The surface mean flow is small and westward (-0.02 m/s) from altimetry line 34 and from in situ data (line 104 cf Table 1 in the Annex).

[Figure]

Figure 2: 30-day low-passed filtered altimetry derived EKE at the mooring array location. The horizontal black dashed line indicates the mean, the blue line marks the value of the mean plus 1.5 standard deviation. The vertical red lines show the period of the mooring deployment.

Figure 7: It would be interesting to superimpose on the Figure 7(a-c) the isopycnals obtained from the moorings to better show the passage of the eddy.

Thank you for your comment and suggestion. Below, is the figure with the isopycnals. Please note that the isopycnals could only be added over the SUMO time series since the other moorings underwent large vertical excursions during the extreme events. We have chosen not to include isopycnals in this figure: isopycnals are derived from vertically interpolated data, which is presented later in the manuscript in Figure 9.

[Figure]

Figure 3: Daily averaged time series of velocity amplitude (m/s) from FLMA (a), FLMB (b) and SUMO (c). The black contours correspond to isopycnals 26.65, 26.85 and 27 kg/m$^3$ computed from the vertically interpolated data. d-e) vertically averaged velocity time series from SUMO (red), FLMA (blue) and FLMB (green) : d) meridional component, e) zonal component. The dashed lines in d-e) are surface geostrophic velocities derived from satellite altimetry co-localised at the mooring locations. X-axis is time (mm/yy). Vertical dashed lines bound A0, A1 and A2' and A2". Black vertical lines correspond to dates considered in Figure 8

Line 224: Is there any direct evidence to show the NIW event was generated by local winds? I cannot find it in the paper. What is the tau signal during each event? It would be good to give estimates of wind energy into NIW, and also try a slab model to demonstrate the wind-driven NIWs.

Thank you for your comment. It was not clear in the manuscript. We rewrote this section as follows: "The wind episodes leading to the NIW generation in the anticyclones were not clearly identified. Winds at OOI are large ($> 8m/s$) with peaks exceeding 15 m/s and there is no clear connection between strong winds and the presence of NIWs at depth at OOI (Figure 12 d). Era-Interim data show various strong wind events ($> 20m/s$) along the eddies´ paths which could have possibly generated the NIWs trapped at depth and observed at OOI. "

Figure 11: It's really hard to see the downward energy propagation of NIWs from the figure? The vertical propagation of NIWs may be estimated directly by fitting a curve to the near-inertial KE maxima (it could also be obtained by fitting the curve to the averaged near-inertial KE values above a threshold value. Tests should be carried out to decide which method is the best).

Thank you for your comment, we agree with you. We modified Figure 11 removing the black dashed lines and including the band-pass filtered KE (see figure below). We modified the text as follows:

"As an example of the wave activity, we show the vertical shear of the horizontal velocity components as A1 crosses the SUMO mooring (Figure 11 a and b). The vertical shear features clear wavy patterns close to the inertial period with vertical wavelength of about 50 m (Figure 11 a and b). The kinetic energy of the band-pass filtered velocities (14-20 h) shows local maxima along isopycnals 26.65 and 27.00 kg/m$^3$ between June 1 and 14 with no obvious connection to the surface (Figure 11 c)."

Vertical propagation is unclear.

[Figure]

Figure 4: a-b) Vertical shear of velocity components for SUMO during event A1. c) 14-20 h band-pass filtered Ke for SUMO during event A1. Contours correspond to isopycnals 26.65, 26.85 and 27 kg/m3 computed from the vertically interpolated data.

---

## Author Comment (AC2)

Summary and recommenndation: The authors analysed multi-year observations from a mooring array deployed in the Argentine Basin, and focued on a few intense anticyclonic eddy events during when near-inertial waves were also excited and mixing was likely to occur. The manuscript is overall clear and well written, and in my opinion, only minor revisions are needed so that the manuscript may be considered for publication.

Thank you for your comments and your time

General comments:

Figure 3: It seems that there are other extreme events occurred around 04/15, 12/16, 03/17, 04/17. Are these also events of mesoscale structures? Could you add some descriptions these events as well?

Indeed, in these dates, the time series show large anomalies. However, the salinity anomalies at 180 m did not exceed the threshold (mean + 3 std) that we established to consider the events as extreme. The SSH maps for those dates also show anticyclonic mesoscale structures. Below are snapshots of the SSH corresponding to the dates when eddies approached the moorings. These events are also observed in the new EKE time series (new Figure 6).

We added the following sentence in the conclusion in line 315:"Indeed, other less extreme anticyclonic eddies crossed OOI moorings: they are seen as peaks in OOI hydrographic data and EKE time series"

[Figure]

Figure 1: a-c) SSH (m) and geostrophic velocities (arrows) from satellite altimetry in April 8 2015, December 31 2016 and March 21 2017.

Figure 6: A large fraction of mesoscale signals would be filtered out when you used the a low-pass window of 180 days. Why don't you use a shorter temporal window like 30 days? Also, is there a mean flow in this region?

Thank you for your comment. We modified the figure using a low-pass window of 30 days (see figure below). We also observe that the year 2016 had the largest EKE values, which lasted for a long time. Indeed, there is a small mean flow. The surface mean flow is small and westward (-0.02 m/s) from altimetry (line 34) and from in situ data (line 104 cf Table 1 in the Annex).

[Figure]

Figure 2: 30-day low-passed filtered altimetry derived EKE at the mooring array location. The horizontal black dashed line indicates the mean, the blue line marks the value of the mean plus 1.5 standard deviation. The vertical red lines show the period of the mooring deployment.

Figure 7: It would be interesting to superimpose on the Figure 7(a-c) the isopycnals obtained from the moorings to better show the passage of the eddy.

Thank you for your comment and suggestion. Below, is the figure with the isopycnals. Please note that the isopycnals could only be added over the SUMO time series since the other moorings underwent large vertical excursions during the extreme events. We have chosen not to include isopycnals in this figure: isopycnals are derived from vertically interpolated data, which is presented later in the manuscript in Figure 9.

[Figure]

Figure 3: Daily averaged time series of velocity amplitude (m/s) from FLMA (a), FLMB (b) and SUMO (c). The black contours correspond to isopycnals 26.65, 26.85 and 27 kg/m$^3$ computed from the vertically interpolated data. d-e) vertically averaged velocity time series from SUMO (red), FLMA (blue) and FLMB (green) : d) meridional component, e) zonal component. The dashed lines in d-e) are surface geostrophic velocities derived from satellite altimetry co-localised at the mooring locations. X-axis is time (mm/yy). Vertical dashed lines bound A0, A1 and A2' and A2". Black vertical lines correspond to dates considered in Figure 8

Line 224: Is there any direct evidence to show the NIW event was generated by local winds? I cannot find it in the paper. What is the tau signal during each event? It would be good to give estimates of wind energy into NIW, and also try a slab model to demonstrate the wind-driven NIWs.

Thank you for your comment. It was not clear in the manuscript. We rewrote this section as follows: "NIWs are generated by a variety of mechanisms, including winds, nonlinear interactions with waves of other frequencies, lee waves over bottom topography, and geostrophic adjustment; the partition among these is not known, although the wind is likely the most important (e.g., Alford et al. 2016). The mechanism at the origin of the trapped NIWs at depth was not identified. There is no connection between strong winds and the presence of NIWs at depth at OOI (Figure 12 d). "

Figure 11: It's really hard to see the downward energy propagation of NIWs from the figure? The vertical propagation of NIWs may be estimated directly by fitting a curve to the near-inertial KE maxima (it could also be obtained by fitting the curve to the averaged near-inertial KE values above a threshold value. Tests should be carried out to decide which method is the best).

Thank you for your comment, we agree with you. We modified Figure 11 removing the black dashed lines and added the band-pass filtered KE (see figure below). We also added ticks at the inertial period following reviewer 2 suggestion. We modified the text as follows:

"As an example of the wave activity, we show the vertical shear of the horizontal velocity components as A1 crosses the SUMO mooring (Figure 11 a and b). The vertical shear features clear wavy patterns close to the inertial period with vertical wavelength of about 50 m (Figure 11 a and b). The kinetic energy of the band-pass filtered velocities (14-20 h) shows local maxima along isopycnals 26.65 and 27.00 kg/m$^3$ between June 1 and 14 with no obvious connection to the surface (Figure 11 c)."

Vertical propagation is unclear.

[Figure]

Figure 4: a-b) Vertical shear of velocity components for SUMO during event A1. c) 14-20 h band-pass filtered Kinetic Energy for SUMO during event A1. Contours correspond to isopycnals 26.65, 26.85 and 27 kg/m$^3$ computed from the vertically interpolated data. Top x-ticks indicate inertial periods from 0:00 30/5/2016 and bottom x-ticks days in dd/mm/y.

---

## Author Comment (AC3)

In this study, the researchers examined data from a mooring array deployed in the Argentine Basin over several years. They specifically investigated intense anticyclonic eddy events accompanied by the excitation of near-inertial waves and the likelihood of mixing.

The paper demonstrates good writing quality, but there are a few aspects and minor reviews that the authors should revisit to improve the manuscript and be published.
Thank you for your time and your suggestions

Line 12: The citation (Artana et al., 2021) appears to be inappropriate. There exist classical papers demonstrating the intricacy of water masses in this particular region.
Thank you for your comment. We changed the reference to:
Reid, J. L., W. D. Nowlin Jr., and W. C. Patzert, On the characteristics and circulation of the southwestern Atlantic Ocean, J. Phys. Oceanogr., 7, 62-91, 1977

Lines 15 and 16: Citations concerning the hydrodynamics of the Brazil Current are absent in this context.
Thank you, we added the following references:
Peterson, R. G., and L. Stramma (1991), Upper-level circulation in the South Atlantic Ocean, Prog. Oceanogr., 26, 1–73, doi:10.1016/0079-6611(91)90006-8.
Gordon, A. L., and C. L. Greengrove (1986), Geostrophic circulation of the Brazil-Falkland Confluence, Deep Sea Res., Part A, 33, 573–585, doi:10.1016/0198-0149(86)90054-3.

Line 57: What is the bin size of the ADCP? It is essential to provide this information explicitly as you are calculating vertical shear using these measurements.
Thank you for your comment. The OOI ADCP data is distributed with a vertical resolution of 10 m and a total of 52 bins. This is now explicitly mentioned in text (line 58).

Line 63-65: This sentence might be improved. Please, revisit the cited paper and provide more details about the data processing.
Artana et al., 2020 is a data processing report. We added further information about the data processing:
"The data processing of the physical variables is detailed in Artana et al. (2020). We removed spikes from temperature and salinity. Measurement quality of the mooring sensors, assessed by comparison with shipboard measurements (CTD profiles and bottle measurements) taken near

the moorings during deployment and recovery cruises, was nominal (accuracy of 0.02 psu for salinity and 0.002°C for temperature)."

Line 67-68: "Data quality ... close to the surface" Same as the comment above. How did you quantify the data quality?

Thank you for your comment. Indeed, the sentence was confusing, we were referring to the missing data. We modified the sentence as follows:

"ADCP from mooring A (FLMA) and the surface mooring (SUMO) did not work during the third deployment (Table 2). Gaps in the upper layer occurred when FLMB and FLMA underwent vertical excursions due to strong currents. Data return was almost 100%, except for periods of mooring draw-downs and for velocity measurements close to the surface. Spiky and noisy patterns were often found in winter, concurrent with strong winds sometimes down to 150 m. They were identified and discarded."

Fig3: While I concur with the authors' observation that the periods of A0, A1, and A2 could be associated with mesoscale features, it is worth noting the presence of numerous similar signals. Do these signals also have a connection to mesoscale eddies? A discussion exploring this aspect would enhance our comprehension of these figures.

Thank you for your comment. We agree with you. This point was also raised by reviewer 1. Other peaks are observed in the hydrographic time series, such as those on 04/15, 12/16, and 03/17, although they are not as extreme as those corresponding to A0, A1 and A2. These peaks are also associated with anticyclonic mesoscale structures. Below we show SSH snapshot for April 8 2015, December 31 2016 and March 21 2017 as examples. They show anticyclonic mesoscale structures close to the mooring array. We added the following comment in the conclusion: "Indeed, other less extreme anticyclonic eddies crossed OOI moorings: they are seen as peaks in OOI hydrographic data and EKE time series".

[Figure]

Figure 1: a-c) SSH (m) and geostrophic velocities (arrows) from satellite altimetry in April 8 2015, December 31 2016 and March 21 2017.

Line 118: "... significant trend.." Has a statistical time-series analysis been conducted, or is there a citation supporting this statement? If not, it is advisable to refrain from using the term "significant," as it implies a requirement for a statistical approach.

Thank you for your comment, the trend significance was tested with a Mann Kendall test. We added this information in the text as follows:
"A 29-year-long surface EKE time series was derived from satellite altimetry at the array site (Figure 6). The 29 year-long EKE time series does not feature any significant trend (Mann Kendall test applied). Year 2016 stood out with EKE values of 1000 x $10^{-4}$ m$^2$/s$^2$ (Figure 6)."

Line 160: "..was originated from the Brazil Current" propagated from the Brazil Current domain.

Thank you, we modified

Line 164: As the analysis does not encompass the investigation of the generation mechanism of these eddies, it is advisable to avoid using the term "generation."

Thank you, we agree. We modified the sentence as follows and avoid the word "generation" in the rest of the manuscript when referring to eddies:
"A0 and A1 were first detected in a region (40°S, 36°W) characterized by a strong gradient in planetary potential vorticity associated with a seamount (Figure 1a)."

Fig10: I apologize for any oversight in the manuscript. Please review the document to ensure that the definitions of CW and CCW are provided.

Thank you. Indeed, the definitions were missing. We added the definitions in the figure caption. Thank you.

Line 218: significant ¿ proeminent ???
Thank you, we agree. Prominent is more appropriate. We modified.

Line 220: The authors mentioned the possibility of local wind as a driving force for the generation of NIWs. However, I am unclear about the basis for this speculation. While it is true that NIWs are predominantly influenced by wind, I recommend conducting additional analysis to support this claim. One potential approach could involve utilizing a simple mixed-layer slab model, which would greatly enhance the final version of the paper.

Thank you for your comment. The wind analysis was not clear in the manuscript. We rewrote this section as follows:
"NIWs are generated by a variety of mechanisms, including winds, nonlinear interactions with waves of other frequencies, lee waves over bottom topography, and geostrophic adjustment; the partition among these is not known, although the wind is likely the most important (e.g., Alford et al. 2016). The mechanism at the origin of the trapped NIWs at depth was not identified. There is no connection between strong winds and the presence of NIWs at depth at OOI (Figure 12 d). "

Lines 228-230: It seems that you are referring to the zeta-refraction mechanism. In light of

this, a discussion about this process is warranted to further elaborate on its role and significance.

Thank you, we modified the paragraph as follows:

Theoretical studies have shown that near-inertial waves interact with the vorticity of mesoscale structures and can be trapped by them (Kunze, 1995). The background vertical relative vorticity shifts the inertial frequency (f) to the effective inertial frequency (feff) through the zeta-refraction mechanism (e.g. Mooers, 1975; Thomas et al., 2020; Rama et al., 2022). Indeed, the horizontal velocity shear of anticyclonic eddies shifts f to a lower "effective planetary vorticity" ($f_{eff}$ = f +$\zeta$/2) (Kunze, 1985). In the southern hemisphere (f<0), positive $\zeta$ of anticyclonic eddies result in $| f_{eff} | < | f |$ in their cores. As a consequence of the lowering of $| f |$ in anticylonic eddies the NIWs become superinertial and their vertical group velocity increases (e.g. Kunze et al., 1995). When NIWs are excited inside a region of anticyclonic relative vorticity, they can have frequencies below f (red shift) and thus remain trapped as they cannot propagate out of the rotating region. As the waves propagate downward from the surface, they reflect off the sides of the anticyclonic region and then stall within a critical layer at the base of the anticyclonic vorticity region, where the group velocity drops to 0. This results in an accumulation of wave energy in a critical layer following tilted isopycnals, and, eventually part of the energy is dissipated by buoyancy release through vertical mixing (e.g. Kunze, 1985; Kunze et al., 1995; Martinez-Marrero et al., 2017; Kawaguchi et al., 2020). Using the $\zeta$ values reported in Table 4, $f_{eff}$ approximately corresponds to a period of about 18 h, consistent with the red shifts observed in Figure 10 spectra.

Figure 11: It would be more advantageous to represent the x-axis as an inertial period rather than using real dates. This adjustment may provide better clarity on the behavior of NIWs. Thank you for your suggestion. Indeed, it improves the figure information. We added ticks in the x axis indicating the inertial period. Following reviewer 1 comment we also added the band-pass filtered KE.

[Figure]

Figure 2: a-b) Vertical shear of velocity components for SUMO during event A1. c) 14-20 h band-pass filtered Kinetic Energy for SUMO during event A1. Contours correspond to isopycnals 26.65, 26.85 and 27 kg/m$^3$ computed from the vertically interpolated data. Top x-ticks indicate inertial periods from 0:00 30/5/2016 and bottom x-ticks days in dd/mm/y.

Line 246 and Figure 12c: Positive values of $\zeta/f$ in the Southern Hemisphere should be negative. Please check if you are computing properly.

Indeed. Thank you. We modified the figure:

[Figure]

Figure 3: a) Envelope of the 14-20 h band-pass filtered shear intensity at SUMO. Black isolines are 3 isopycnals computed from SUMO vertically interpolated density time series: 26.65, 26.85, 27.00 kg/m³. b) Satellite altimetry derived $\zeta/f$ (Left y-axis) and temperature at 350 m at SUMO (Right y-axis) c) Envelope of the 14-20 h band-pass filtered density at SUMO. d) Wind intensity from Era-Interim at SUMO. Red colors indicate values larger than 10 m/s.

Item 4.3: Is the vertical scale of measurements (dz) sufficient to suggest that vertical shear can potentially overcome stratification and destabilize the flow? A comprehensive discussion on this topic would enhance the manuscript. Additionally, it is mentioned that the authors obtained values of Ri close to a quarter using a coarse spatial resolution by calculating the minimum ratio between $N^2$ and $S^2$. Could you provide insight into the data distribution? Is the minimum ratio a representative measure of Ri? I recommend comparing it with a bulk estimation of Ri and assessing whether using the minimum Ri value adequately captures the

overall representation.

Thank you for your comment. Ri estimate is tentative. Please note that N2 resolution is limited as N2 is derived from vertically interpolated data. The discrete measurement levels are indicated with white dots in the y-axis of Figure 12. The Ri values are closer to 1. This is discussed in the text:

"Though the existence of a universal critical Richardson number $Ri_c$ at which turbulent mixing can start is still debated, a lower bound Ri $< Ri_c$ is often assumed, for example $Ri_c = 1/4$, originally derived from linear stability of steady stratified shear flows (Miles, 1961), or $Ri_c = 1$ from non-linear stability analysis (e.g Abarbanel et al., 1984)"

Below we show the estimation of Ri considering the hourly data (not the daily minimum). The minimum Ri captures the overall representation.

We now comment on the limitation of the Ri computation in the manuscript:

"The Richardson number has been considered as a valuable proxy for turbulence production in the oceans and the presence of low fine-scale Ri associated with internal waves (NIWs) has been frequently observed in the ocean interior (e.g. Martinez Marrero 2019). In these studies, it was assumed that shear occurring at scales beyond the resolution of the ADCP would sufficiently reduce the total Richardson number to trigger shear instability. The Ri estimates, based on a relatively coarse vertical resolution of N2 and S2, show low values close to 1 in the specific regions where trapped NIWs were detected. Indeed, the four anticyclonic features show a low Ri at depth (between 300 and 400 m for A0 and A1 and between 400 and 500 m for A2' and A2") at the limit between SACW and AAIW (indicated with a white isoline in Figure 13). The low Ri suggests that favourable conditions for enhanced mixing between AAIW and SACW occurred at depth within the anticyclones where NIWs are trapped and eventually break."

[Figure]

Figure 4: a-c) Brunt-Väisälä frequency ($N^2$) (hourly averaged data are linearly interpolated in the vertical), amplitude of vertical shear and Richardson Number during event A0 at mooring B. The Richardson number was tentatively computed as the daily minimum of the ratio between $N^2$ and the vertical shear of the horizontal velocity. The black isolines are the 26.65 and 26.85 kg/m$^3$ isopycnals and the white isoline is the 27.00 kg/m$^3$ isopycnal which corresponds to the limit between SACW and AAIW. White dots in the y-axis in (a) indicate the discrete measurement levels.

d-f) Same as (a-c) at the surface mooring during events A0, A1 and A2. g-i) Same as (a-c) at mooring A during event A2 event. j-l) Same as (a-c) at mooring B during A2 event.

Label of section 5: Conclusions¿ Summary and conclusions We agree, thank you.

Line 300: Rossby baroclinic deformation radius. Please provide the actual value.
Thank you. The deformation radius at OOI is about 25 km. We added in the text: They had radii of about 40 km A2' probably smaller and A2" slightly larger and a vertical extent of about 2500 m. In the region the Rossby baroclinic radius is $\sim$ 25 km (Chelton et al.,1998).

Line 306: birthplace??? Could you really access the origin of such an eddy?

Thank you, we agree with you. We modified the sentence paragraph as follows:

Where do those anticyclonic structures come from? Satellite altimetry suggested that the three eddies had different origins and paths. A1 detached from STF 600 km to the northeast of the array over a topographic rise (40°S, 36°W) 60 days before crossing the OOI array. A2' was tracked back as a large anticyclonic eddy detached from the Brazil Current overshoot (43°S, 52°W). It travelled during 4 months with the Zapiola Anticyclonic Circulation before reaching the array. While at the array, satellite altimetry maps suggest that A2' merged with A2" which arrived after a 16-day trip from its first-detected place at 42°S, 40°W. After a time life of 3, 6 and 2 months for A1, A2' and A2" respectively, the three eddies were reabsorbed by the STF.

Line 318-319: The sentence in question appears to be overly speculative. Please rephrase it. Thank you for your comment, we removed the sentence.

---

## Author Response (AR2)

Public justification (visible to the public if the article is accepted and published): Thank you for responding carefully to the suggestions of both reviewers. I am happy to accept the paper subject to some minor technical changes:

Thank you very much for your time and your swift answer.

First, as noted by the production team, please use one of the online colour-blind-friendly figure checkers to ensure that your figures will be accessible to all readers.
Thank you. We modified colors in Figure 2, Figure 3, Figure 5, Figure 7 and Figure A1. We carefully checked with the color-blind simulator that figures are accessible to all readers.

Second, your figure 4 correctly has no units for salinity as you are (presumably) using EOS-80 rather than TEOS-10? Please could you add in the methods section that your salinities will be presented on the practical salinity scale, to avoid confusion for future readers? I note that there are a few instances where the unit "psu" has been included, which is not approved by EOS-80. Please could you remove these psu units?
Thank you, we remove "psu" in all figures and in the manuscript and stated in the method section that salinities are presented on the practical salinity scale.